



**Synthetic ozone deposition and stomatal uptake at flux tower sites**
Jason A. Ducker[1], Christopher D. Holmes[1], Trevor F. Keenan[2,3], Silvano Fares[4], Allen H.
Goldstein[3], Ivan Mammarella[5], J. William Munger[6], Jordan Schnell[7]
[1] Department of Earth, Ocean, and Atmospheric Science, Florida State University, Tallahassee,
Florida
[2] Lawrence Berkeley National Laboratory, University of California, Berkeley, California
[3] Department of Environmental Science, Policy, and Management, University of California,
Berkeley, California
[4] Council of Agricultural Research and Economics (CREA), Research Centre for Forestry and
Wood, Arezzo, Italy.
[5] Institute for Atmosphere and Earth System Research/Physics, PO Box 68, Faculty of Science,
University of Helsinki, Finland
[6] Department of Earth and Planetary Sciences, Northwestern University, Evanston, Illinois
[7] NOAA Geophysical Fluid Dynamics Laboratory, Princeton, New Jersey
**Abstract**
We develop and evaluate a method to estimate $O_3$ deposition and stomatal $O_3$ uptake across
networks of eddy covariance flux tower sites where $O_3$ concentrations and $O_3$ fluxes have not
been measured. The method combines standard micrometeorological flux measurements, which
constrain $O_3$ deposition velocity and stomatal conductance, with a gridded dataset of observed
surface $O_3$ concentrations. Measurement errors are propagated through all calculations to
quantify $O_3$ flux uncertainties. We evaluate the method at three sites with $O_3$ flux measurements:
Harvard Forest, Blodgett Forest, and Hyytiälä Forest. The method reproduces 83% or more of
the variability in daily stomatal uptake at these sites with modest mean bias (21% or less). At
least 95% of daily average values agree with measurements within a factor of two and, according
to the error analysis, the residual differences from measured $O_3$ fluxes are consistent with the
uncertainty in the underlying measurements.
The product, called synthetic $O_3$ flux or SynFlux, includes 43 FLUXNET sites in the United
States and 60 sites in Europe, totaling 926 site-years of data. This dataset, which is now public,
dramatically expands the number and types of sites where $O_3$ fluxes can be used for ecosystem
impact studies and evaluation of air quality and climate models. Across these sites, the mean
stomatal conductance and $O_3$ deposition velocity is 0.03-1.0 cm s$^{-1}$. The stomatal $O_3$ flux during
the growing season (April-September) is 0.5-11.0 nmol m$^{-2}$ s$^{-1}$ with a mean of 4.5 nmol m$^{-2}$ s$^{-1}$
and the largest fluxes generally occur where stomatal conductance is high, rather than where $O_3$
concentrations are high. The conductance differences across sites can be explained by
atmospheric humidity, soil moisture, vegetation type, irrigation, and land management. These



stomatal fluxes suggest that ambient $O_3$ degrades biomass production and $CO_2$ sequestration by
20-24% at crop sites, 6-29% at deciduous broadleaf forests, and 4-20% at evergreen needleleaf
forests in the United States and Europe.
**1  Introduction**
Surface ozone ($O_3$) is toxic to both people and plants. Present-day and recent historical $O_3$ levels
reduce carbon sequestration in the biosphere (Reich and Lassoie, 1984; Guidi et al., 2001; Sitch
et al., 2007; Ainsworth et al., 2012), perturb the terrestrial water cycle (Lombardozzi et al., 2012,
2015), and cause around $25 billion in annual crop losses (Reich and Amundson, 1985; Van
Dingenen et al., 2009; Avnery et al., 2011; Tai et al., 2014). The basic plant responses to $O_3$
injury are well established from controlled exposure experiments (e.g. Wittig et al., 2009;
Ainsworth et al., 2005, 2012; Hoshika et al., 2015) but few datasets are available to quantify $O_3$
fluxes and responses for whole ecosystems or plant functional types that are represented within
regional and global biosphere and climate models. The eddy covariance method has been widely
used to measure land-atmosphere fluxes of carbon, water, and energy and evaluate their
representation in models (Baldocchi et al., 2001; Bonan et al., 2011), but few towers measure $O_3$
fluxes (Munger et al., 1996; Fowler et al., 2001; Keronen et al., 2003; Gerosa et al., 2004;
Lamaud et al., 2009; Fares et al., 2010; Stella et al., 2014; Zona et al., 2014). A recent review
identified just 78 field measurements of $O_3$ fluxes over vegetation during the last 4 decades,
many lasting just a few weeks (Silva and Heald, 2017). This paper demonstrates a reliable
method to estimate $O_3$ fluxes at 103 eddy covariance flux towers spanning over two decades to
enable $O_3$ impact studies on ecosystem scales.
The land surface is a terminal sink for atmospheric $O_3$ due to the reactivity of $O_3$ with
unsaturated organic molecules and the modest solubility of $O_3$ in water. Surface deposition
removes about 20% of tropospheric $O_3$, making it an important control on air pollution (Wu et
al., 2007; Young et al., 2013, Kavassalis and Murphy, 2017). This $O_3$ deposition flux includes
stomatal uptake into leaves, where $O_3$ can cause internal oxidative damage, and less harmful
non-stomatal deposition to plant cuticles, stems, bark, soil, and standing water (Fuhrer, 2000;
Zhang et al., 2002; Ainsworth et al., 2012). $O_3$ can also react with biogenic volatile organic
compounds in the plant canopy air and this process is commonly included in non-stomatal
deposition (Kurpius and Goldstein, 2003). The deposition flux (mol m$^{-2}$ s$^{-1}$) can be described as:
$$F_{O_3} = v_d n(\chi - \chi_0) = v_d n\chi \qquad (1)$$
where $\chi$ and $\chi_0$ are the $O_3$ mole fractions (mol mol$^{-1}$) in the atmosphere and at the surface,
respectively, $n$ is the molar density of air (mol m$^{-3}$), and $v_d$ is a deposition velocity (m s$^{-1}$) that
expresses the net vertical $O_3$ transport between the height where $\chi$ is measured and the surface.
$F_{O_3}$ is defined positive for flux towards the ground. Eq. 1 reasonably assumes that $\chi_0 = 0$ because
terrestrial surfaces have abundant organic compounds that react with and destroy $O_3$. The
deposition velocity can be decomposed into resistances (s m$^{-1}$) for aerodynamic transport ($r_a$),





diffusion in the quasi-laminar layer ($r_b$), stomatal uptake ($r_s$), and non-stomatal deposition ($r_{ns}$)
(Wesely, 1989):

$$v_d^{-1} = r_a + r_b + (r_s^{-1} + r_{ns}^{-1})^{-1}. \tag{2}$$

For stomatal and non-stomatal processes, the rates are often expressed as conductances (m s$^{-1}$),
which are the inverse of the resistances: $g_s = r_s^{-1}$ and $g_{ns} = r_{ns}^{-1}$. The sum of stomatal and non-
stomatal conductances is the vegetation canopy conductance, $g_c = g_s + g_{ns}$. The stomatal $O_3$
flux is the portion of $F_{O_3}$ that enters the stomata, and can be described as:

$$F_{s,O_3} = F_{O_3} g_s (g_s + g_{ns})^{-1} = v_d n \chi g_s (g_s + g_{ns})^{-1}. \tag{3}$$

To construct the synthetic $O_3$ flux, or SynFlux, we use measurements of $O_3$ concentration and
standard eddy covariance flux measurements to derive nearly all of the terms in Eqs. 1-3 from
surface observations, using minimal additional information from remote sensing and models.
This enables the estimation of $F_{O_3}$ and $F_{s,O_3}$, as described in Section 2. In Section 3 we then
evaluate the method against observations at three sites that measure $F_{O_3}$, examine the importance
of stomatal and non-stomatal deposition, and compare flux-based metrics of $O_3$ damage with
concentration-based metrics. Finally, we discuss the strengths, limitations, and implications of
our approach in Section 4.
**2    Data sources and methods**
**2.1    SynFlux: synthetic $O_3$ flux**
The FLUXNET2015 dataset (Pastorello et al., 2017) aggregates measurements of land-
atmosphere fluxes of $CO_2$, $H_2O$, momentum, and heat at sites around the world
(http://fluxnet.fluxdata.org/data/fluxnet2015-dataset, accessed 24 February 2017). Measurements
are made with the eddy covariance method on towers above vegetation canopies (Baldocchi et
al., 2001; Anderson et al., 1984; Goldstein et al., 2000) with consistent gap-filling (Reichstein et
al., 2005; Vuichard and Papale, 2015) and quality control across sites (Pastorello et al., 2014).
Flux and meteorological quantities are reported in half hour intervals. We analyze data from all
sites in the United States and Europe in the FLUXNET2015 Tier 1 dataset. This analysis is
restricted to the US and Europe because these regions have dense $O_3$ monitoring networks,
described below. There are 103 sites meeting these criteria, all listed in Table S1 with references
to full site descriptions. Three of these sites—Blodgett Forest, Harvard Forest, and Hyytiälä
Forest—measure $O_3$ flux with the eddy covariance method, which we will use in Sect. 3 to
evaluate our methods.
SynFlux aims to tightly constrain $O_3$ deposition resistances using measured water, heat and
momentum fluxes, in contrast to other methods that rely more heavily on atmospheric models or
standard meteorology observations (Finkelstein et al., 2000; Mills et al. 2011; Schwede et al.,
2011; Yue et al., 2014). From the eddy covariance measurements, we derive the resistance





components of Eq. 2 using methods similar to past studies (Kurpius and Goldstein, 2003; Gerosa
et al., 2005; Fares et al., 2010). The aerodynamic and quasi-laminar layer resistances ($r_a$ and $r_b$,
respectively) are derived from measured wind speed, friction velocity, and fluxes of sensible and
latent heat every half hour using Monin-Obukhov similarity theory (Foken, 2017). The stomatal
conductance for $O_3$ ($g_s$) is derived from the measured water vapor flux and meteorological data
every half hour with the inverted Penman-Monteith equation (Monteith, 1981; Gerosa et al.,
2007). Some studies instead calculate $g_s$ from the measured gross primary productivity (GPP)
(Lamaud et al., 2009; El-Madany et al., 2017). That method likely underestimates the stomatal
flux, however, because the $g_s$/GPP ratio increases as humidity rises and because $g_s$ remains non-
zero when GPP has ceased at night (Dawson et al., 2007; Medlyn et al., 2011). Appendix A
provides further details of these calculations. To avoid complications to the Penman-Monteith
equation from wet canopies, we exclude times when dew may be present (RH > 80%), and days
with precipitation (> 5mm). We also exclude the top and bottom 1% of $g_s$ values, which include
many unrealistic outliers (e.g. $|g_s| > 0.5$ m s$^{-1}$). Figure 1 shows the mean stomatal conductance
during the growing season (April-September) at all sites.
The terms in Eqs. 1-3 that cannot be derived from FLUXNET2015 measurements are $O_3$ mole
fraction and non-stomatal conductance. The $O_3$ mole fraction is taken from a gridded dataset of
hourly $O_3$ measurements that spans the contiguous United States and Europe (Schnell et al.,
2014). This dataset has 1° spatial resolution, so some differences from measured $O_3$ abundances
at individual sites are inevitable. Schnell et al. (2014) estimated these errors to be 6-9 ppb (rms)
or about 15% of summer mean $O_3$ in the US and similar in Europe. Figure 2 shows that the
daytime gridded $O_3$ concentrations correlate well with observations at three flux tower sites
where $O_3$ was measured ($R^2 = 0.63$-$0.87$) and have modest negative bias (5-10 ppb, –12 to –
28%), consistent with the accuracy reported by Schnell et al. (2014). We use the Zhang et al.
(2003) parameterization of non-stomatal conductance, which accounts for $O_3$ deposition to leaf
cuticles and ground and has been evaluated at sites in North America. The parameterization
requires leaf-area index, which we take from satellite remote sensing (Claverie et al., 2014;
2016), snow depth, which we take from MERRA2 reanalysis (GMAO, 2015; Gelaro et al.,
2017), and standard meteorological data provided by FLUXNET2015.
Figure 3 shows the stomatal $O_3$ flux at each site calculated with Eq. 3, then averaged over the
April-September growing season. Figure S1 shows the corresponding total $O_3$ flux (Eq. 1). We
refer to these products as the "synthetic" total and stomatal $O_3$ fluxes ($F'_{O_3}$ and $F'_{s,O_3}$,
respectively) and use a prime to distinguish them from the measured $O_3$ fluxes ($F_{O_3}$ and $F_{s,O_3}$)
that are only available at a few sites. Together, we refer to $F'_{O_3}$ and $F'_{s,O_3}$ as SynFlux. In total, the
measurements required to calculate $F'_{s,O_3}$ are $O_3$ mole fraction, sensible and latent heat fluxes,
friction velocity, temperature, pressure, humidity, canopy height, and leaf area index. There are
43 sites in the US and 60 sites in Europe within the FLUXNET Tier 1 database with sufficient
measurements to calculate $F'_{s,O_3}$.





**2.2    Observed O$_3$ flux**
We evaluate SynFlux and its inputs at three sites where O$_3$ flux measurements are available:
Harvard Forest, Massachusetts, United States (Munger et al., 1996); Blodgett Forest, California,
United States (Fares et al., 2010); and Hyytiälä Forest, Finland (Keronen et al., 2003;
Mammarella et al., 2007; Rannik et al., 2009). These forest sites sample a range of
environmental and ecosystem conditions summarized in Table 1. All three sites have at least 6
years of half-hourly or hourly flux measurements. Two sites are evergreen needleleaf forests
(Blodgett and Hyytiälä), while one is a deciduous broadleaf forest containing some scattered
stands of evergreen needleleaf trees (Harvard). Climate also differs across these sites. Blodgett
Forest has a Mediterranean climate with cool, wet winters and hot, dry summers. Hyytiälä and
Harvard Forests have cold winters and wetter summers, with Harvard Forest being the warmer of
the two.
Harvard Forest water vapor flux measurements were recalibrated for this work based on
matching water vapor mixing ratio measured by the flux sensor to levels calculated from ambient
relative humidity and air temperature, resulting in a 30% increase in evapotranspiration during
the 1990s and no change since 2006. In addition, we remove sub-canopy evaporation from the
measured water vapor flux before the Penman-Monteith calculation. Based on past
measurements at these sites, the sub-canopy fraction of evapotranspiration is 20% at Hyytiälä
Forest, 10% at Harvard Forest in summer, and 50% at Harvard Forest in months without leaves
(Moore et al., 1996; Launiainen et al., 2005). We are unable to make this correction at all
FLUXNET sites since water vapor flux is typically measured only above canopy.
At these three sites, observed $v_d$, $g_{ns}$, and $F_{s,O_3}$ can be derived from the $F_{O_3}$ measurements with
methods that differ slightly from Sect. 2.1. O$_3$ deposition velocity is inferred from measurements
of O$_3$ concentration and flux via $v_d = F_{O_3}(n\chi)^{-1}$. Resistance or conductance terms $r_a$, $r_b$, and $g_s$
are calculated as described in Sect. 2.1, then both canopy and non-stomatal conductance are
derived from observations via $g_c = (v_d^{-1} - r_a - r_b)^{-1}$ and $g_{ns} = g_c - g_s$, respectively. With
those values, Eq. 3 gives the observed stomatal O$_3$ flux. The half-hourly or hourly measured and
synthetic flux still have some outliers (Fig. S2), but the error analysis reveals that most of the
outlying points have large uncertainties.
**2.3    Gap filling for friction velocity**
The FLUXNET2015 dataset uses gap filling for most flux and meteorological measurements
(Vuichard and Papale, 2015), but not for friction velocity ($u_*$), which is required to calculate $v_d$



and $F'_{s,O_3}$. Filling this one variable would significantly reduce the fraction of missing data in our
analysis. Monin-Obukhov similarity theory predicts that friction velocity is proportional to wind
speed in the surface layer, for a given roughness length and stability regime (Foken, 2017). On
this basis, we regress the available friction velocity measurements against wind speed and net
radiation (a proxy for stability) separately for each site and month (a proxy for vegetation
roughness). This gap filling was possible at 91 sites that report net radiation measurements.

The predicted friction velocities from the regression model are highly correlated with available
observations ($R^2 > 0.7$) and have minimal mean bias (±10%) at 68 out of 91 eligible sites (Fig.
S3). At the remaining 23 sites, frequent stagnant and stable conditions ($u_* \lesssim 0.5$ m s$^{-1}$) degrade
the regression performance. We used the regression model to fill missing friction velocity
measurements and were thus able to increase the number of $F'_{s,O_3}$ estimates by 1-20%. The
differences between monthly mean $F'_{s,O_3}$ with and without gap filling are 10% (rms), so although
the $u_*$ gap filling is a potential source of uncertainty, the $F'_{s,O_3}$ estimates are robust.

**2.4    Error analysis, averaging, and numerical methods**

We quantify the errors in $F'_{O_3}$, $F'_{s,O_3}$, and all other calculated variables from the measurement
uncertainties using standard techniques for propagation of errors through all equations (see
Appendix B). This method provides the uncertainty, quantified as standard deviation, of each
variable in each half hour interval. The error analysis reveals that $F'_{s,O_3}$ and other derived
quantities have uncertainties that change from hour to hour by two orders of magnitude (Fig. S2).
In addition, many extreme values of $F'_{s,O_3}$, $g_s$, and other variables have very large uncertainties.
We retain these outliers in our analysis and use the error analysis to appropriately reduce their
influence on averages and other statistics, as described below, without discarding data.

The FLUXNET2015 dataset contains error estimates for sensible and latent heat measurements.
We use these reported values in the error analysis. Where uncertainties in these fluxes are
missing, we fill the gaps using a linear regression of available flux errors against flux values for
that site. For friction velocity, the uncertainty is the prediction error in the linear model used for
gap filling (Sect. 2.3). Based on expert judgment, the standard deviation of $O_3$ mole fraction is
set to 20%, pressure to 0.5 hPa, temperature to 0.5 K, relative humidity to 5%, and canopy height
to the lesser of 15% or 2 m. For leaf area index, we use reported uncertainties in the remote
sensing for each plant functional type (Claverie et al., 2013; 2016). The Zhang et al. (2003) $g_{ns}$
parameterization has 5 vegetation-specific parameters and all are assigned 50% standard
deviation. Zero error is assumed for the flux tower height. Based on these inputs, the median
relative uncertainty in $F'_{s,O_3}$ is 44%, but it rises to several hundred percent for some half-hour
intervals. The error analysis shows that most of the uncertainty in $F'_{s,O_3}$ derives from uncertainty
in the latent heat flux measurement.





Daily and monthly averages of $F'_{s,O_3}$ and other quantities are constructed in stages. We first
calculate a mean diurnal cycle for the day or month by pooling measurements during each hour
in a maximum likelihood estimate, a weighted average that accounts for the uncertainty in each
measurement. The maximum likelihood estimate is appropriate when combining values from the
same distribution, which is expected to apply for measurements within a particular hour, but not
across hours of the day. We then average across hours with an unweighted mean to calculate the
daily or monthly value. Seasonal values are the unweighted mean of the months they contain.
Uncertainties are propagated through each stage of these averages, as detailed in Appendix B.
Our discussion focuses on daily averages of daytime data when the sun is at least 4° above the
horizon.
Analyses are performed in Python 3.5 with NumPy, Pandas, PySolar, and Statsmodels (Reda and
Andreas, 2005; Van Der Walt et al., 2006; McKinney, 2010; Seabold et al., 2010). We quantify
the slope and strength of linear relationships between variables using standard major axis fitting
(SMA, Warton et al., 2006), the non-parametric Thiel-Sen slope (Sen, 1968), and coefficient of
determination ($R^2$).

### 2.5    Data availability

The SynFlux dataset produced in this work is available in the supplementary information for
download and use. The dataset includes synthetic stomatal and total $O_3$ fluxes, $O_3$ concentrations,
$O_3$ deposition velocity, canopy conductance, stomatal conductance, and all of their propagated
uncertainties. Monthly mean values are provided, with and without $u_*$ gap filling, for 103 sites
totaling 926 site-years.

## 3    Results and discussion

### 3.1    Evaluation of synthetic fluxes

Figure 4 compares daily daytime averages of synthetic $F'_{s,O_3}$ to measured $F_{s,O_3}$. At all three sites,
$F'_{s,O_3}$ is strongly correlated with measured values ($R^2$ = 0.83-0.93). The mean and median biases
are −16 to −21% and at least 95% of $F'_{s,O_3}$ values agree with measurements within a factor of 2.
The majority of $F'_{s,O_3}$ values lie near the 1:1 line with $F_{s,O_3}$ and the slopes (0.71 to 0.85) reflect
this. For 98% of points, the differences between $F'_{s,O_3}$ and $F_{s,O_3}$ are less than the 95% confidence
interval derived from the error analysis (two-sided t test). Thus, the errors in $F'_{s,O_3}$ are consistent
with the uncertainty in the observations. The half hourly $F'_{s,O_3}$ values perform similarly well
against observations (Fig. S4), but our analysis focuses on averages. Overall, these results





suggest that synthetic $F'_{s,O_3}$ is a reliable estimate of stomatal $O_3$ uptake into plants that can be
used at eddy covariance sites without $O_3$ measurements.
The measurements enable us to evaluate synthetic total deposition, $F'_{O_3}$, as well, although this is
less relevant to ecosystem impacts than stomatal uptake, $F'_{s,O_3}$. Figure S5 shows that bias (–13 to
+65%), slope (0.3-1.4), and $R^2$ (0.05-0.43) for $F'_{O_3}$ are all worse than for $F'_{s,O_3}$. The reasons can
be derived from Eq. 3. The canopy resistance for $O_3$ is normally much greater than the quasi-
laminar layer and aerodynamic resistances, meaning $r_c \gg r_a$ and $r_c \gg r_b$, often by a factor of 3-
10. Therefore, the $O_3$ deposition velocity is approximately $v_d \approx r_c^{-1} = g_c$. Under these
conditions, Eq. 1 simplifies to $F_{O_3} \approx n\chi(g_s + g_{ns})$ and Eq. 3 simplifies to $F_{s,O_3} \approx n\chi g_s$. While
$g_s$ is calculated from measured $H_2O$ fluxes, $g_{ns}$ comes from a parameterization, which inevitably
introduces error into $g_{ns}$ and $F'_{O_3}$. Since $F'_{s,O_3}$ has little sensitivity to $g_{ns}$ or its errors, it can be
calculated more accurately than $F'_{O_3}$, as seen when comparing Figures 4 and S4. Despite these
larger errors, the $F'_{O_3}$ mean is within 50% of the observed value at two sites and within a factor of
2 at all, which may be useful for some applications, given the paucity of prior $F_{O_3}$ measurements.
**3.2     Stomatal and non-stomatal deposition**
Figure 5 shows the seasonal cycles of observed $O_3$ deposition velocity and its important
components at the three study sites with $O_3$ flux measurements. For low or moderately reactive
gases like $O_3$, canopy resistance is typically greater than aerodynamic or quasi-laminar layer
resistance, so it controls the overall deposition velocity. At these three sites, deposition velocity
is lowest in winter (0.1-0.2 cm s$^{-1}$) and highest in summer (0.5–0.6 cm s$^{-1}$). Stomatal
conductance, which peaks when weather conditions favor growth, explains most of this seasonal
variation, except at Blodgett Forest as discussed below. Stomatal conductance is generally
thought to exceed non-stomatal conductance during the growing season at most vegetated sites
(Wesely, 1989; Zhang et al., 2003). At both Harvard and Hyytiälä Forests, the mean stomatal
conductance (0.2-0.6 cm s$^{-1}$) is 1.5-6 times larger than non-stomatal conductance (0.08-0.2 cm s$^{-1}$)
$^{-1}$) during the growing season, so about 60-90% of $O_3$ deposition occurs through stomatal uptake.
In winter at these sites, the calculated stomatal conductance can exceed canopy conductance,
which is not possible, but is likely an artifact of evaporation from soil or snow exceeding leaf
transpiration at that time of year. At Blodgett, non-stomatal conductance slightly exceeds
stomatal conductance in summer (0.4 vs. 0.3 cm s$^{-1}$). The fast non-stomatal deposition is
explained by $O_3$ reacting with biogenic terpenoid emissions below the flux measurement height
(Kurpius and Goldstein, 2003; Fares et al., 2010). These biogenic emissions depend strongly on
temperature and light and have a large seasonal cycle with maxima in summer and minima in
winter, so stomatal uptake is generally < 50% of $O_3$ deposition at Blodgett in the summer but >
70% in winter.






A recent analysis of $O_3$ flux measurements at Harvard Forest suggests that non-stomatal
deposition averages 40% of daytime $O_3$ deposition during summer months, with a range of 10-
60% across years (Clifton et al., 2017). Our analysis of the same site, using re-calibrated latent
heat flux measurements, does not support such a large role for non-stomatal deposition at this
site in summer. As seen in Fig. 5, only 15% of $O_3$ deposition is non-stomatal during these
months, with a range of 4-32% across years. At Hyytiälä Forest, our results are consistent with
prior work that found that the non-stomatal deposition is 26% to 44% of daytime $O_3$ deposition
during the growing season (Rannik et al., 2012). Nevertheless, non-stomatal deposition equals or
exceeds stomatal uptake where there are large terpene emissions (e.g. Blodgett) and at some
other temperate sites that probably lack large biogenic emissions (Fowler et al., 2001; Cieslik,
2004; Lamaud et al., 2009; Stella et al., 2011; El-Madany et al., 2017).

At Harvard and Hyytiälä Forests, the parameterized $g_{ns}$ has a similar mean to measurements
during summer, with discrepancies less than a factor of two (Fig. 5). The observed day-to-day
variability in $g_{ns}$ is as large as the variability in $g_s$ at Harvard and Hyytiälä Forests and the
calculated $g_{ns}$ does not reproduce it ($R^2 < 0.09$), so an important but undetermined non-stomatal
process is missing from the parameterization. At Blodgett Forest, the parameterized $g_{ns}$ is one-
third of measured $g_{ns}$ in summer, but this is not surprising since the parameterization does not
account for $O_3$ reactions with biogenic volatile organic compounds (BVOC), which are known to
be important at this site (Fares et al., 2010). We attempted, unsuccessfully, to use BVOC
emissions from the MEGAN biogenic emission model (Guenther et al., 2012) to improve the $g_{ns}$
parameterization, but the correlations between measured $g_{ns}$ and compounds that react fastest
with $O_3$ (monoterpenes and sesquiterpenes) were poor ($R^2 \leq 0.15$). On that basis, synthetic $F'_{O_3}$
may also underestimate total $O_3$ deposition at other sites with high monoterpene and
sesquiterpene emissions, such as warm-weather pine forests, but synthetic $F'_{s,O_3}$ should retain its
quality everywhere.

**3.3   Spatial patterns of synthetic fluxes**

Across the 43 sites in the US shown in Fig. 3, mean $F'_{s,O_3}$ during the growing season ranges from
0.5 to 11.0 nmol m$^{-2}$ s$^{-1}$ with an average of 4.4 nmol m$^{-2}$ s$^{-1}$. The highest $F'_{s,O_3}$ generally occurs in
the Midwest (5-9 nmol m$^{-2}$ s$^{-1}$ in Wisconsin, Michigan, Nebraska, Ohio) due to its moderate $O_3$
concentrations (Fig. S6) and moisture levels, which promotes stomatal conductance (Fig. 1). The
Western US has higher average $O_3$ concentrations, but generally lower moisture and stomatal
conductance, especially the Southwest US, so $F'_{s,O_3}$ (0-4 nmol m$^{-2}$ s$^{-1}$) is mostly lower than the
Midwest. Land cover, land management, and plant types drive large differences in $F'_{s,O_3}$ between
nearby sites, even when $O_3$ concentrations and meteorology are similar. For example, three



Nebraska sites are all crop fields and $O_3$ concentrations are nearly identical, but two irrigated
fields have higher stomatal conductance and higher $F'_{s,O_3}$ than the nearby rainfed field (6.2 vs.
4.8 nmol m$^{-2}$ s$^{-1}$). Two sites in central California have high $g_s$ and $F'_{s,O_3}$ compared to surrounding
sites due to irrigation and naturally wet soil in the California Delta. A combination of topography
and climate is also an important factor in California: forest sites in the Sierra Nevada mountains
have lower $g_s$ and $F'_{s,O_3}$ than the lowland crops and wetland grasses. In Oregon, an evergreen
needleleaf site regrowing after a fire has higher $g_s$ and $F'_{s,O_3}$ than two older forest stands nearby.
The differences between 9 Wisconsin forest sites, however, are mostly due to different years of
data at each site combined with interannual variability in $F'_{s,O_3}$; fluxes at these sites are similar in
overlapping years.
Variability across the 60 sites in Europe is controlled by similar factors. Stomatal uptake ranges
from 1.4 to 9.6 nmol m$^{-2}$ s$^{-1}$, with an average of 4.7 nmol m$^{-2}$ s$^{-1}$ (Fig. 3). The Mediterranean
region has high $O_3$ concentrations (Fig. S6), but generally low stomatal conductance due to the
dry climate (Fig. 1). Within this region, vegetation type explains broad patterns. Shrub sites in
Spain, France, and Sardinia have very low $g_s$ (~0.15 cm s$^{-1}$) so $F'_{s,O_3}$ is low (1-3 nmol m$^{-2}$ s$^{-1}$),
while the most of the sites in mainland Italy are broadleaf and evergreen forests that have slightly
greater $g_s$ (~0.2-0.4 cm s$^{-1}$) and $F'_{s,O_3}$ (3-6 nmol m$^{-2}$ s$^{-1}$), despite similar climate and $O_3$. In
central and northern Europe, temperate climate promotes higher stomatal conductance while $O_3$
concentrations remain modest throughout the growing season. The largest $F'_{s,O_3}$ is 9.8 nmol m$^{-2}$ s$^{-}$
$^1$ at a deciduous broadleaf forest in Switzerland, while nearby evergreen forests, cereal crops, and
grasslands all have lower fluxes (6-8 nmol m$^{-2}$ s$^{-1}$). While Finland has generally low $F'_{s,O_3}$ of 2-5
nmol m$^{-2}$ s$^{-1}$, the high end of this range is similar to rural sites in Germany, illustrating that $O_3$
can impact ecosystems with low $O_3$ concentrations far from major industrial emissions.
Table 2 quantifies SynFlux, $O_3$ deposition velocity, and conductance for each plant functional
type. Wetlands, crops, and forests have the highest average $F'_{s,O_3}$, which is about two times
higher than woody savanna or shrublands, the vegetation types with the lowest $F'_{s,O_3}$. The
vegetation types rank in the same order for stomatal conductance, again showing stomata as the
main control on $O_3$ deposition and uptake. Stomatal uptake exceeds non-stomatal uptake for all
plant functional types except woody savanna and shrubland. $O_3$ deposition velocities reported in
the table fall within the ranges of past literature, as reviewed by Silva and Heald (2017).
However, while Silva and Heald found that the mean deposition velocity was greater over
deciduous forests than coniferous forests, crops, or grass, we do not. Rather, we find that
variability between sites within each of these categories is large, having a standard deviation
about 30% of the multi-site mean.
**3.4   Metrics for $O_3$ damage to plants**





Since $O_3$ injures plants mainly by internal oxidative damage after entering the leaves through
stomata, the most physiological predictor of plant injuries is the cumulative uptake of $O_3$ (CUO,
Reich, 1987; Fuhrer, 2000; Karlsson et al., 2004; Cieslik, 2004; Matyssek et al., 2007). CUO is
defined as the cumulative stomatal $O_3$ flux exceeding a threshold flux Y that can be detoxified by
the plant, integrated over a period of time:
$$\mathrm{CUOY} = \sum_i H(F_{s,O_3,i} - Y)(F_{s,O_3,i} - Y)\,\Delta t_i.$$
Here, $H(x)$ is the Heaviside step function and $\Delta t_i$ is the time elapsed during measurement of
$F_{s,O_3,i}$. The sum is carried out over time $i$ in the growing season, which we take to be April to
September. The detoxification threshold varies across vegetation types, even among related
species (Karlsson et al., 2004, Büker et al., 2015), and thresholds for specific FLUXNET sites
are generally unknown. As a compromise, we calculate CUO, with Y=0, and also CUO3, with Y
= 3 nmol m$^{-2}$ s$^{-1}$, which has been suggested as a reasonable generic threshold (Mills et al., 2011).
CUO is always greater than CUO3, but the sites with high CUO tend to also have high CUO3, so
their spatial patterns are similar (Fig. S7).
While CUO is a physiological dose, concentration-based metrics remain common for assessing
ozone impacts because they are easier to measure. Concentration-based metrics quantify $O_3$ in
ambient air irrespective of whether that $O_3$ enters leaves. These metrics follow the general form
$$M = \sum_i w(\chi_i)\,(\chi_i - \chi_c)\,\Delta t_i$$
where $w(\chi)$ is a weighting function applied to the $O_3$ mole fraction $\chi$, and $\chi_c$ is a constant. Like
CUO, the sum is usually over time $i$ during the growing season. Three of the most common
concentration-based $O_3$ metrics are the mean $O_3$ concentration, the accumulated concentration
over a threshold of 40 ppb (AOT40; UNECE, 2004), and the sigmoidal-weighted index (W126;
Lefohn and Runeckles, 1987). For mean, $w(\chi) = (\sum \Delta t_i)^{-1}$ and $\chi_c = 0$. For AOT40, $w(\chi) =$
$H(\chi - \chi_c)$ and $\chi_c = 40$ ppb. For W126, $w(\chi) = (1 + 4403 \exp(-(126\ \mathrm{ppb}^{-1})\chi))^{-1}$ and $\chi_c =$
0. Both AOT40 and W126 use only daytime (8am-8pm) measurements and W126 also takes the
maximum value over a 3-month period. The weighting functions for AOT40 and W126 give
little or no weight to $O_3$ concentrations below 40 ppb. In addition, W126 gives increasing weight
to concentrations up to about 110 ppb and full weight for higher concentrations based on the
understanding that exposure to high $O_3$ concentrations is more injurious than moderate or low
concentrations. Other concentration-based metrics (e.g. SUM60) use other thresholds or
weighting functions, but many are strongly correlated with AOT40 or W126 or otherwise
qualitatively similar (Paoletti et al., 2007).
The spatial patterns of AOT40 and W126 closely resemble that of mean $O_3$ concentration in the
US and Europe despite their different weighting functions (Fig. S7). The CUO and CUO3 spatial
patterns, however, are similar to $F'_{s,O_3}$ and distinct from the concentration-based metrics. This



illustrates that locations with high AOT40 or W126, like the Southwest US or Mediterranean
Europe, can have low CUO.
Even though concentration-based metrics do not measure the physiological $O_3$ dose to plants,
they can be useful if the metric is proportional to the flux-based dose and injuries. Indeed, many
controlled experiments and observational studies have documented correlations between both
AOT40 and W126 and either uptake or plant injuries (e.g. Fuhrer et al., 1997; Cieslik, 2004;
Musselman et al., 2006; Matyssek et al., 2010). However, many of these studies were carried out
at a single site or under conditions where stomatal conductance was relatively steady while $O_3$
concentrations varied, for example by maintaining well-watered soil. When stomatal
conductance varies widely, such as between arid and humid climates or seasons, concentration-
based metrics may not correlate with stomatal $O_3$ flux (Mills et al., 2011).
Figure 6 shows that all of the concentration-based metrics are poorly correlated with CUO across
the sites (AOT40: $R^2 = 0.05$, W126: $R^2 = 0.03$, mean $O_3$: $R^2 = 0.04$). Humidity helps explain some
of the scatter in Figure 6. The sites with high concentration-based metrics and low CUO have
high vapor pressure deficit (VPD), low stomatal conductance, and are mostly in the western US
and Mediterranean Europe. Restricting the analysis to humid sites (VPD < 1.5 kPa) does not
improve the correlation ($R^2 \approx 0.05$) and at the arid sites (VPD > 1.6 kPa) the concentration-based
metrics are modestly anti-correlated with CUO (AOT40: $R^2 = 0.19$, W126: $R^2 = 0.05$, mean $O_3$:
$R^2 = 0.37$). This result reinforces that concentration-based metrics can misrepresent CUO and
plant injuries (Mills et al., 2011).
From the CUO values in Table 2, we can estimate the range of $O_3$ impacts on biomass
production at the FLUXNET sites. Although species vary in their sensitivity to $O_3$, several
studies suggest that the biomass production of broadleaf and needleleaf trees decreases 0.2 to 1%
per mmol $m^{-2}$ of CUO (Karlsson et al., 2004; Wittig et al., 2007; Hoshika et al., 2015).
Combining the mean CUO for each plant functional type (Table 2) with these sensitivities, our
work implies that $O_3$ reduces the biomass production at these FLUXNET sites by 6-29% for
deciduous broadleaf forests and 4-20% for needleleaf forests. The range represents the spread of
reported dose-response sensitivities within each plant type, meaning the least and most $O_3$-
sensitive species. Lombardozzi et al. (2013) caution that species-specific responses to $O_3$ may
not generalize to plant functional types, but the biomass reductions calculated here still indicate
the general magnitude of expected $O_3$ damages. Several broadleaf crops are more sensitive to $O_3$,
with biomass reductions of 1.3-1.6% per mmol $m^{-2}$ of CUO3 (Mills et al., 2011). That sensitivity
implies 20-24% drop in biomass production at FLUXNET crop sites. Some studies have
quantified $O_3$ dose-response relationships with other thresholds Y = 1.6 to 6 nmol $m^{-2}$ $s^{-1}$ (e.g.
Karlsson et al., 2007; Pleijel et al., 2004, 2014), but the sensitivities have similar magnitude.
Fares et al. (2013) also demonstrated 12-19% reduction in gross primary production due to $O_3$ at
some of the same crop and forest FLUXNET sites. Using prognostic models of $O_3$





concentrations and stomatal uptake, several past studies have also suggested that $O_3$ reduces
biomass production and $CO_2$ sequestration by 4-20% in the US and Europe (Sitch et al., 2007;
Wittig et al., 2007; Mills et al., 2011; Yue et al., 2014, 2016; Lombardozzi et al., 2015). Our
results support this range of impacts, although some FLUXNET sites and species likely
experience greater $O_3$ injury, but here the CUO is highly constrained from observations and
therefore avoids the additional uncertainties of atmosphere-biosphere models.
**4   Conclusions**
We have demonstrated a method to estimate $O_3$ fluxes and stomatal $O_3$ uptake at eddy
covariance flux towers wherever regional $O_3$ monitors exist. The method, called SynFlux,
derives stomatal conductance and $O_3$ deposition velocity from standard eddy covariance
measurements and combines them with gridded $O_3$ concentrations from air quality monitoring
networks. We apply this method to the FLUXNET2015 dataset and derive synthetic flux
estimates at 43 sites in the United States and 60 sites in Europe, totaling 926 site-years of
observations. $O_3$ deposition measurements have previously only been sporadically available for a
few sites around the world, so this work dramatically increases the flux data available for
understanding $O_3$ impacts on vegetation and for evaluating air quality and climate models.
Three sites with long-term $O_3$ flux measurements provide an independent test of SynFlux. These
comparisons show that daily averages of synthetic stomatal $F'_{s,O_3}$ correlate well with measured
$F_{s,O_3}$ ($R^2 = 0.83$-$0.93$) and have a mean bias under 22% at all sites. At all three sites 95% of the
synthetic $F'_{s,O_3}$ values differ from measurements by a factor of 2 or less. The differences between
$F'_{s,O_3}$ and $F_{s,O_3}$ are also consistent with propagated uncertainty in the underlying measurements.
Synthetic total deposition, $F'_{O_3}$, is sensitive to errors in the parameterized non-stomatal
conductance, but mean values are still with a factor of 2 of observations. The errors in this
dataset are modest compared with differences between observations and regional and global
atmospheric chemistry models that are frequently a factor of 2 or more (Zhang et al., 2003;
Hardacre et al., 2015; Clifton et al., 2017; Silva and Heald, 2017), illustrating the utility of this
dataset for evaluating models and $O_3$ impacts.
Across flux tower sites in the US and Europe, $F'_{s,O_3}$ ranges from 0.5 to 11.0 nmol m$^{-2}$ s$^{-1}$ during
the summer growing season. The spatial pattern of $F'_{s,O_3}$ is mainly controlled by stomatal
conductance rather than $O_3$ concentration. Patterns of stomatal conductance and $F'_{s,O_3}$ in turn are
explained by climate, especially atmospheric and soil moisture, vegetation types, and land
management, such as irrigation. $O_3$ concentration-based metrics (AOT40, W126, mean $O_3$) have
been widely used to evaluate $O_3$ damages to plants because they are easier and cheaper to
measure than the cumulative uptake of $O_3$ (CUO) into leaves. However, these metrics have very

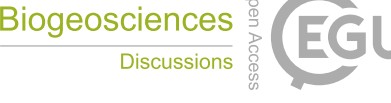



little correlation with CUO ($R^2 \leq 0.05$) across FLUXNET sites. Using dose-response
relationships between CUO and biomass reduction, we estimate that $O_3$ reduces biomass
production and carbon uptake by 4-29%, depending on the site and plant type. Unlike most past
estimates, which have used prognostic models of $O_3$ uptake, our assessment of biomass reduction
is based on $O_3$ fluxes that are tightly constrained by observations. To promote further
applications in ecosystem monitoring and modeling, the SynFlux dataset is publicly available in
the supplement as monthly averages of $F'_{s,O_3}$, $F'_{O_3}$, $O_3$ deposition velocity, stomatal conductance,
and related variables.



**Acknowledgments**
This work was supported by the Winchester Fund and by the Council on Research Creativity at
Florida State University. Eddy covariance data used here were acquired and shared by the
FLUXNET community, including the AmeriFlux and CarboEuropeIP networks. The FLUXNET
eddy covariance data processing and harmonization was carried out by the European Fluxes
Database Cluster, AmeriFlux Management Project, and Fluxdata project of FLUXNET, with the
support of CDIAC and ICOS Ecosystem Thematic Center, and the OzFlux, ChinaFlux and
AsiaFlux offices. TFK was supported by the Director, Office of Science, Office of Biological
and Environmental Research of the US Department of Energy under Contract DE-AC02-
05CH11231 as part of the RUBISCO SFA. The $O_3$ concentration and flux measurements from
Harvard Forest used in this analysis were supported by the National Science Foundation through
the LTER program and various programs under the U. S. Department of Energy Office of
Science (BER). At Hyytiälä Forest, $O_3$ concentrations and flux measurements were supported by
ICOS-Finland (281255) and Academy of Finland Center of Excellence programme (307331). At
Blodgett Forest, $O_3$ concentrations and flux measurements were supported by ICOS-Finland
(281255) and Academy of Finland Center of Excellence programme (307331). The long term $O_3$
concentration and flux measurements from Blodgett Forest used in this analysis were supported
by a combination of grants from the Kearney Foundation of Soil Science, the University of
California Agricultural Experiment Station, and the U.S. Department of Energy Office of
Science (BER), the National Science Foundation Atmospheric Chemistry Program, and the
California Air Resources Board.





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



Table 1. Description of sites that measure $O_3$ flux and their daytime growing season (April-
September) conditions [a]

|  | Blodgett Forest, California, USA | Hyytiälä Forest, Finland | Harvard Forest, Massachusetts, USA |
|---|---|---|---|
| Latitude, Longitude | 38.8953, –120.6328 | 61.8475, 24.2950 | 42.5378, –72.1715 |
| Plant functional type | Evergreen needleleaf | Evergreen needleleaf | Deciduous broadleaf |
| Years of data | 2001-2007 | 2007-2012 | 1993-1999 |
| Days of observations | 1281 | 1098 | 1281 |
| Canopy height, m | 8 | 15 | 24 |
| GPP, $\mu mol\ m^{-2}\ s^{-1}$ | 9.22 ± 3.55 | 11.1 ± 5.02 | 12.4 ± 7.62 |
| ET, $mmol\ m^{-2}\ s^{-1}$ | 3.25 ± 1.23 | 1.71 ± 0.82 | 2.95 ± 1.70 |
| PAR, $\mu mol\ m^{-2}\ s^{-1}$ | 875 ± 149 | 690 ± 203 | 876 ± 222 |
| Air Temperature, °C | 19.1 ± 5.36 | 13.3 ± 5.99 | 17.65± 5.75 |
| VPD, kPa | 1.51 ± 0.61 | 0.73 ± 0.32 | 0.90 ± 0.34 |
| $O_3$, ppb | 55.4 ± 13.4 | 32.2 ± 8.68 | 48.8 ± 15.8 |
| $F_{s,O_3}$, $nmol\ m^{-2}\ s^{-1}$ | 5.18 ± 2.11 | 4.35 ± 1.66 | 7.23 ± 4.87 |
| Precipitation, $mm\ day^{-1}$ | 0.09 ± 0.49 | 0.42 ± 0.89 | 0.28 ± 0.82 |

[a] Values are mean ± standard deviation of daily averages, using daytime observations only. GPP is gross
primary productivity. ET is evapotranspiration. PAR is photosynthetically active radiation. VPD is vapor
pressure deficit. $F_{s,O_3}$ is observed stomatal $O_3$ flux.
Table 2. Mean $O_3$ SynFlux, deposition velocity and its conductance components during daytime
in the growing season (April-September), grouped by plant functional type (PFT).[a]

| PFT[b] | Sites | Site-Years | $g_s$ | $g_{ns}$ | $g_c$ | $v_d$ | $F'_{O_3}$ | $F'_{s,O_3}$ | CUO | CUO3 |
|---|---|---|---|---|---|---|---|---|---|---|
| CRO | 18 | 148 | 0.42±0.17 | 0.28±0.09 | 0.68±0.18 | 0.53±0.12 | 7.66±1.96 | 4.77±1.52 | 24.8±12.4 | 14.9±9.3 |
| ENF | 25 | 254 | 0.37±0.10 | 0.25±0.06 | 0.60±0.11 | 0.54±0.10 | 7.37±1.33 | 4.61±1.16 | 20.0±5.69 | 11.9±6.30 |
| EBF | 3 | 31 | 0.21±0.02 | 0.15±0.02 | 0.36±0.03 | 0.33±0.03 | 5.02±0.65 | 2.90±0.28 | 12.1±0.81 | 5.12±0.45 |
| DBF | 16 | 158 | 0.41±0.14 | 0.20±0.09 | 0.60±0.18 | 0.53±0.15 | 7.87±2.28 | 5.37±1.69 | 28.6±13.8 | 15.7±6.66 |
| MF | 5 | 83 | 0.44±0.17 | 0.19±0.01 | 0.62±0.15 | 0.56±0.14 | 7.82±1.91 | 5.53±2.15 | 24.9±10.5 | 15.9±8.90 |
| WSA | 2 | 25 | 0.10±0.02 | 0.31±0.06 | 0.39±0.04 | 0.36±0.04 | 6.14±0.20 | 1.47±0.31 | 6.46±1.43 | 2.54±1.72 |
| OSH | 4 | 14 | 0.19±0.07 | 0.29±0.10 | 0.47±0.10 | 0.41±0.09 | 5.69±1.33 | 2.23±0.87 | 8.60±3.27 | 2.27±1.54 |
| CSH | 2 | 15 | 0.27±0.11 | 0.29±0.01 | 0.57±0.09 | 0.49±0.05 | 6.78±0.95 | 3.34±1.24 | 14.3±5.30 | 7.62±5.49 |
| GRA | 18 | 136 | 0.40±0.30 | 0.24±0.11 | 0.64±0.26 | 0.47±0.15 | 7.04±7.04 | 4.12±2.45 | 18.3±10.7 | 9.90±6.98 |
| WET | 10 | 53 | 0.48±0.16 | 0.27±0.09 | 0.74±0.21 | 0.58±0.14 | 8.80±2.74 | 5.77±2.08 | 25.1±9.65 | 19.4±15.6 |

[a] Values are the mean ± standard deviation across sites within each PFT. Units for $g_s, g_{ns}, g_c,$ and $v_d$ are cm
$s^{-1}$. Units for $F'_{O_3}$ and $F'_{s,O_3}$ are $nmol\ m^{-2}\ s^{-1}$.
[b] CRO = crop, ENF = evergreen needleleaf forest, EBF = evergreen broadleaf forest, DBF = deciduous
broadleaf forest, MF = mixed forest, WSA = woody savanna, OSH = open shrubland, CSH = closed
shrubland, GRA = grassland, WET = wetland






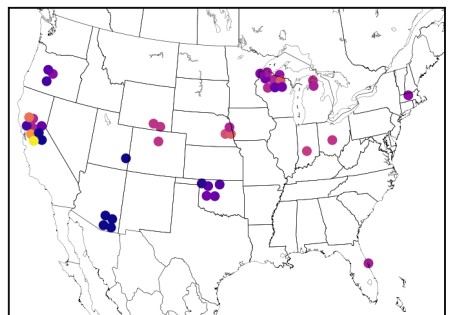 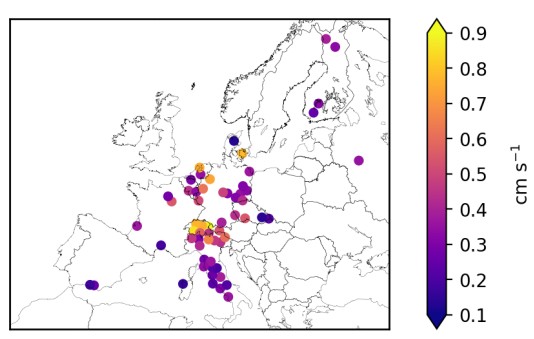

Figure 1. Mean stomatal conductance for $O_3$ ($g_s$) during daytime in the growing season (April-
September) at FLUXNET2015 sites in the United States and Europe. Symbols of some sites have
been moved slightly to reduce overlap and improve legibility.

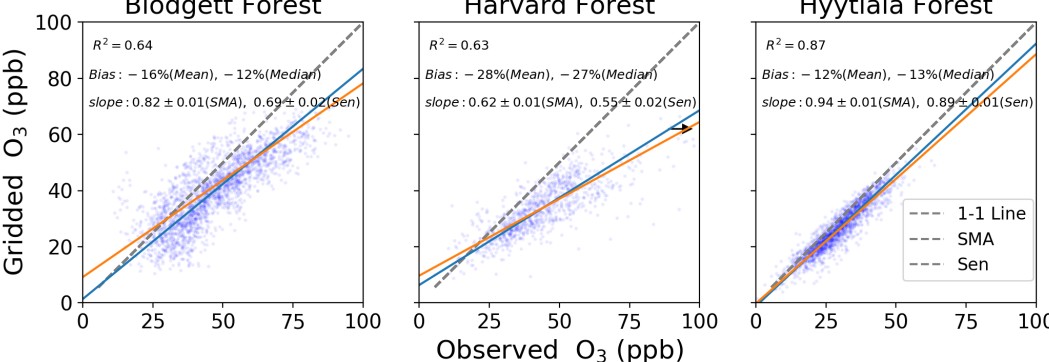

Figure 2. Gridded and observed daily daytime $O_3$ concentrations at Blodgett, Harvard, and
Hyytiälä Forests. Inset numbers provide the coefficient of determination ($R^2$), mean and median
bias, the standard major axis (SMA) slope, the Thiel-Sen (Sen) slope, and the 68% confidence
interval of the slopes. Black arrow points towards outliers that are not shown.




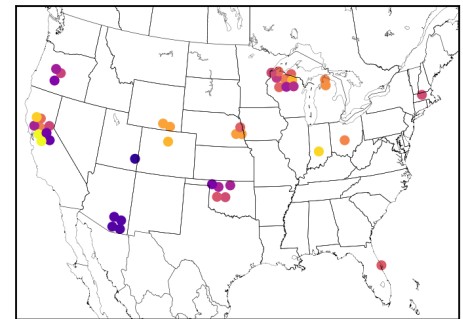 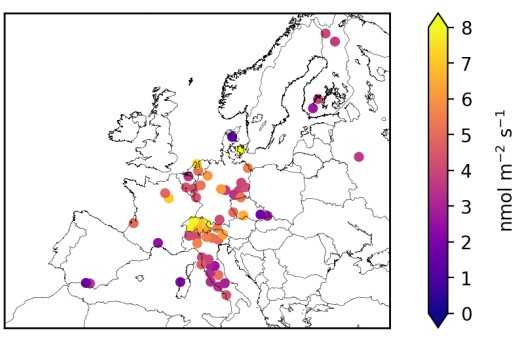

Figure 3. Mean synthetic stomatal O$_3$ flux ($F'_{s,O_3}$, Sect. 2.1) during the daytime growing season
(April-September) at FLUXNET2015 sites in the United States and Europe. Symbols of some
sites have been moved slightly to reduce overlap and improve legibility.

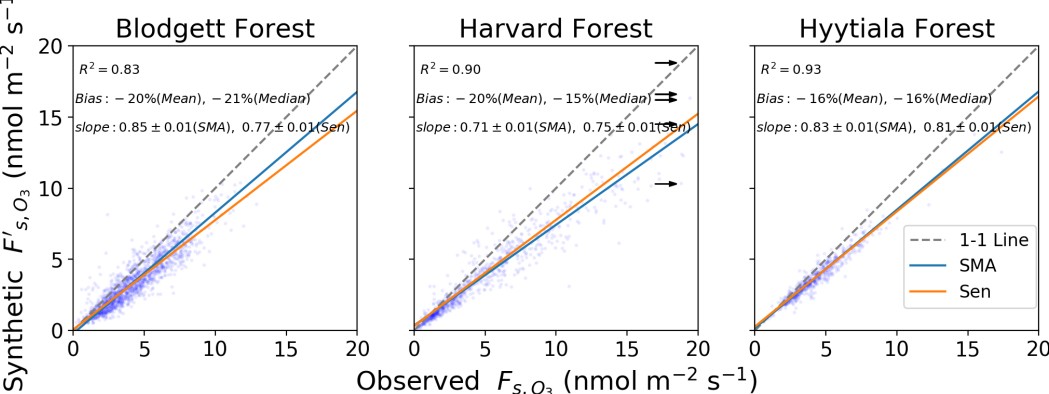

Figure 4. Synthetic and observed daily daytime stomatal O$_3$ flux. See Sect. 2.1 for definition of
$F'_{s,O_3}$ and Fig. 2 for explanation of lines and inset text.

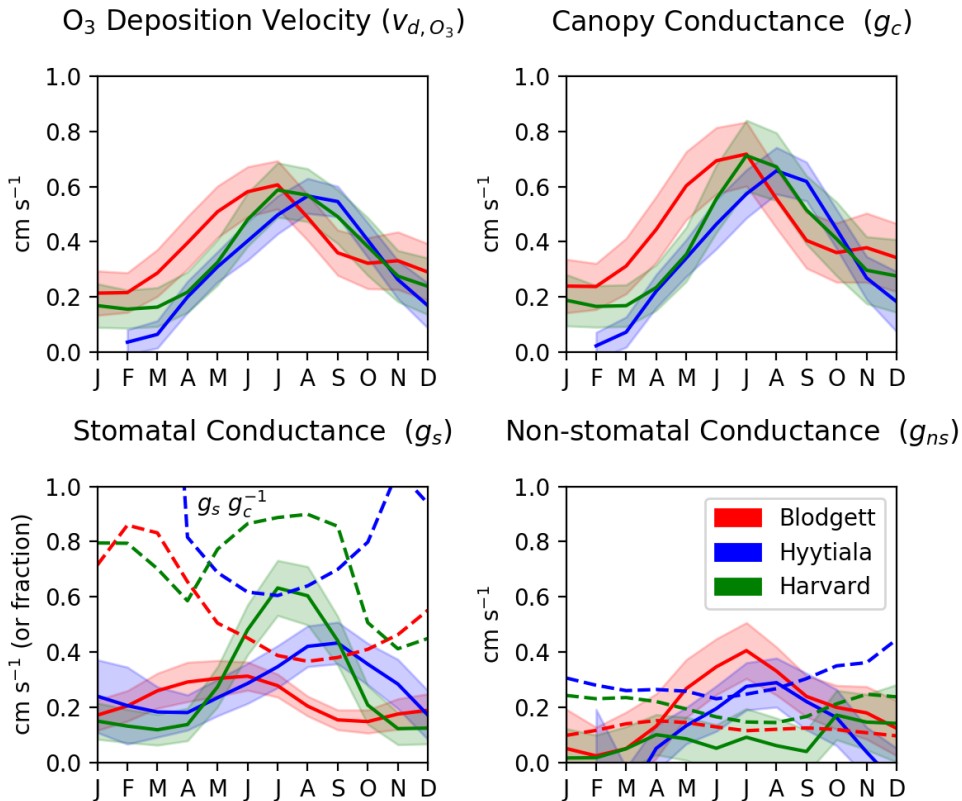

Figure 5. Observed $O_3$ deposition velocity and its in-canopy components at sites with $O_3$ flux
measurements. Lines show means and shaded regions show standard deviation of daily values
for each month. Dashed lines on the stomatal conductance panel show the stomatal fraction of
total canopy conductance ($g_s\, g_c^{-1}$) and dashed lines on the non-stomatal conductance panel show
the parameterized value.



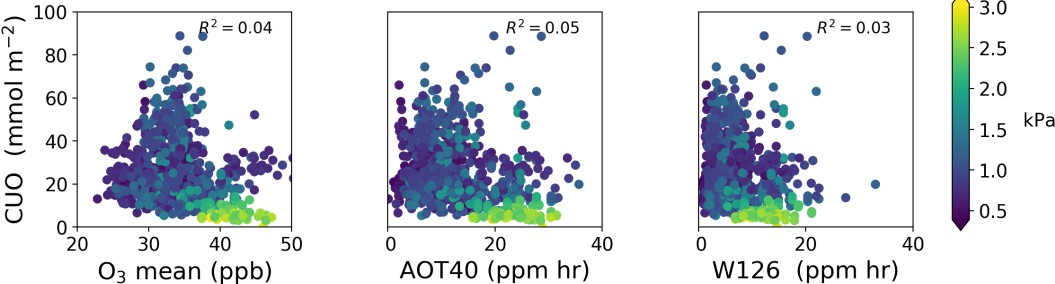

Figure 6. Comparison of cumulative uptake of $O_3$ (CUO) to concentration-based metrics of $O_3$
exposure during the daytime growing season (April-September) at 103 sites: mean $O_3$
concentration (left), AOT40 (center), and W126 (right). There is one value (dot) per site per
year. Colors show mean vapor pressure deficit during the growing season.