# Peer review of "Synthetic ozone deposition and stomatal uptake at flux tower sites"

_Biogeosciences, 2018_

## Short Comment (SC1) · 30 Apr 2018

O. Clifton

oclifton@ldeo.columbia.edu

Ducker et al. (2018) find that their estimates of stomatal conductance suggest that the nonstomatal fraction of ozone dry deposition ranges from 4 to 32% across years at Harvard Forest and note that this is different from what Clifton et al. (2017) find (20 to 58%) (see Ducker et al. (2018) lines 318-323). Stomatal conductance estimates are critical for inferring the nonstomatal fraction of deposition as the nonstomatal conductance is calculated as a residual of the canopy conductance (inferred from ozone eddy covariance fluxes) and the stomatal conductance. Ducker et al. (2018) attribute the difference between our estimates of the year-to-year range in the nonstomatal fraction of deposition to their re-calibration of the water vapor fluxes used in the stomatal conductance estimate. First, I ask them to clarify whether their range of the nonstomatal

fraction is indeed comparable with mine. Second, I use the authors' stomatal conductance estimates (given in their supplementary material) to show below that their estimates of the nonstomatal fraction of deposition are similar to those given by Clifton et al. (2017). I recommend that the authors clarify their discussion of my previous work based on this finding.

First, it is unclear how Ducker et al. (2018) arrive at their reported 4-32% range. When I examine their Figure 5, the figure referenced for these numbers, I can infer the 4-32% range from the error bars, which the caption says represent one standard deviation across daily estimates. If the 4-32% range represents the standard deviation across daily estimates, then these numbers are not directly comparable to mine and the text should be revised accordingly. The 20-58% given in my paper represents the range in the summertime daytime mean nonstomatal conductance across yearly values, not the spread across daily values. If Ducker et al. (2018) actually calculate the mean nonstomatal conductance for each year to obtain their range of 4-32% to compare directly with Clifton et al. (2017), then their approach needs to be more clearly documented in the manuscript.

To investigate whether the re-calibration of water vapor fluxes leads to the differences in the fractions of stomatal (or nonstomatal) deposition in Ducker et al. (2018) vs. Clifton et al. (2017), I downloaded the authors' monthly mean stomatal conductance estimates given in their supplementary material. I divided their summertime (June-September) daytime mean stomatal conductance estimates at Harvard by my own estimates of canopy conductance (9am-4pm June-September for each year). My methods for inferring canopy conductance from the ozone eddy covariance flux measurements at Harvard Forest are described in Clifton et al. (2017), and are similar to those of Ducker et al. (2018). Inferring the canopy conductance depends on estimates of the resistances to turbulence and molecular diffusion. These resistances are typically relatively small during the summer daytime compared to the total resistance to deposition calculated from the ozone eddy covariance fluxes, so there should not be substantial
differences between our estimates of canopy conductance. Dividing their stomatal conductance estimate by my estimate of canopy conductance suggests that summertime mean stomatal deposition varies from ∼50 to 100% of the total deposition during 1993-2000. This corresponds to nonstomatal deposition varying from ∼0 to 50% of the total ozone dry deposition from year to year (see Figure 1 below). This is wider than the range presented by Ducker et al. (2018) (i.e., 4-32%). Given the uncertainties in the measurements and differences in our approaches especially with respect to inferring stomatal conductance, I think it is fair to say that this range is similar to the range in Clifton et al. (2017) of 20-58%. My analysis here suggests that re-calibrated water vapor fluxes are not the root cause of the major differences in the ranges given by Ducker et al. (2018) vs. Clifton et al. (2017). Rather, it seems more likely that the differences reflect consideration of the spread in daily variability (Ducker et al., 2018) rather than the year-to-year range (Clifton et al., 2017).

References. Clifton, O. E., A. M. Fiore, J. W. Munger, S. Malyshev, L. W. Horowitz, E. Shevliakova, F. Paulot, L. T. Murray, and K. L. Griffin (2017). Interannual variability in ozone removal by a temperate deciduous forest, Geophysical Research Letters, 44. https://doi.org/10.1002/2016GL070923

Ducker, J. A., C. D. Holmes, T. F. Keenan, S. Fares, A. H. Goldstein, I. Mammarella, J. W. Munger, and J. A. Schnell (2018). Synthetic ozone deposition and stomatal uptake at flux tower sites, Biogeosciences Discussions. https://doi.org/10.5194/bg-2018-172

Figure 1. Summertime nonstomatal fraction of ozone dry deposition at Harvard Forest for 1993-2000. For the nonstomatal fraction shown in black I use the Ducker et al. (2018) stomatal conductance (gs) estimate and my estimate of canopy conductance (gc) from ozone eddy covariance flux observations. I average their monthly daytime gs (from SynFlux_nofill_day.csv) across June-September. My gc estimates are for June-September 9am-4pm; the methods are described in Clifton et al. (2017) and are similar to Ducker et al. (2018). For the nonstomatal fraction shown in yellow, I use the same estimate of gc as for the black line, but I use my estimate of gs (see Clifton et al. (2017)

[Figure]

methods). The ranges shown here are similar, suggesting that the much lower range of the nonstomatal fraction (4-32%) reported by Ducker et al. (2018) reflects their use of daily variations to infer interannual variability.
* * *
[Figure]

[Figure]

**Fig. 1.**

---

## Referee Comment (RC1) · Anonymous Referee #1 · 9 May 2018

Ducker et al. (2018) develop a large dataset of total and stomatal deposition of ozone using micrometeorological observations at flux towers, process models, satellite data, and a gridded product of surface ozone concentrations from air quality networks. They evaluate the simulated deposition against ozone eddy covariance flux observations at three sites with long-term measurements. Then, the authors use their new dataset of simulated stomatal and total deposition to examine the drivers of spatial variability and estimate ozone damage to plants.

The authors harp on the utility of their "synthetic" ozone deposition dataset. Although I see the value in the stomatal deposition estimates, the authors do not really show that their dataset can tell us anything new. Further, I am not convinced that the total ozone deposition estimates are useful, especially when variability in non-stomatal

deposition is not simulated accurately. I understand the authors need non-stomatal conductance to estimate the stomatal ozone flux, but is a non-stomatal estimate that gets the variability completely wrong better than a constant? After major revisions I think this manuscript will be suitable for publication.

I would like to see a discussion of previous studies that use observed water vapor fluxes from FLUXNET sites to infer stomatal conductance (e.g., Novick et al., 2016, Lin et al., 2018). I would also like to see a discussion concerning the authors' not accounting for the evaporation contribution to the observed water flux at most sites. There have been several recent papers suggesting partitioning methods or ways of estimating evaporation (e.g., Zhou et al., 2016, Gentine et al, 2016). While for many sites transpiration will dominate the observed water vapor flux during the growing season, evaporation will dominate at other times. I'm not sure why the authors even attempt to estimate deposition during these times.

The difference between the "synthetic" and "observed" stomatal fluxes of ozone needs to be clarified in the manuscript. Do the authors use equation (3) to calculate both? If so, which terms are different between the two calculations? I assume $g\_s$ is the same between the calculations, so it seems like $v\_d$, [O3], and $g\_ns$ differ. The authors should include a discussion of the major drivers of differences between the synthetic and inferred stomatal fluxes in Section 3.1. The authors do show the differences in flux-tower and Schnell [O3] in Figure 2, but I think they need to also show and discuss differences in the $v\_d$ and $g\_ns$ that are from the ozone eddy covariance flux observations vs. estimated with $g\_s$ and the Zhang model.

The comment from Olivia Clifton needs to be addressed. In general, a clear understanding of how accurate the authors' estimates are on different timescales is critical to understanding the estimates' usefulness. Going back to Clifton et al. 2017, they show strong inter-annual variation in ozone deposition velocity at the Harvard Forest. Do the authors' estimates capture this variation? What about at Blodgett and Hyytiälä? How much of the variability that is captured with the authors' daily estimates is due to

capturing the seasonality of vegetation (i.e., LAI, drought) rather than inter-annual and daily variability?

Terminology 1) I find the abbreviations used in this study to be very un-intuitive. In the very least I ask that the authors change F_s,O3 to F_sto,O3 so that the "s" can't be confused with "synthetic".

2) The stomatal conductance, stomatal ozone fluxes, and g_ns should never be referred to as "observed". I understand that the authors need to distinguish between their synthetic fluxes and the quantities that are inferred from ozone eddy covariance flux observations, but it is misleading to call the latter "observed".

3) The units for stomatal fluxes are given as nmol m-2 s-1. Is this nmol O3 m-2 s-1?

Line-by-line comments

Line 67: It's not exactly true that deposition removes 20% of tropospheric ozone. Deposition is 20% of the total loss.

Line 71-73: The authors should elaborate here on the types of ambient reactions that matter for measured ozone fluxes. For example, does reaction of isoprene and ozone in the canopy matter?

Line 92: I think "minimal additional information from remote sensing and models" is a stretch. The entire synthetic non-stomatal estimate is from a model that relies on remote sensing. If non-stomatal deposition were a minimal fractional of the total, the phrasing would be ok. However, numerous studies have suggested it is not.

Line 118-119: Again, I take issue with this statement. Yes, the authors' work uses stomatal conductance inferred from observations, which may be better than parameterized stomatal conductance, but the authors are also reliant on modeling and standard meteorology observations for their non-stomatal deposition estimates.

Line 127-130: I find the authors' argument against using GPP to indicate g_s a bit

flawed. First, not all GPP-based g_s estimates predict g_s as a linear function of GPP. Second, the authors are not incorporating nighttime g_s into this study, so why does the point about nighttime GPP matter?

Line 147: To my knowledge, the Zhang et al. (2002) parameterization has not been evaluated at sites in North America. Rather Zhang et al. (2002) build their model using ozone fluxes from a couple of sites in the eastern US. This non-stomatal parameterization is rather uncertain (eg., see discussions in Wolfe et al., 2011, Stella et al., 2011, Altimir et al., 2006).

Line 148-150: A brief analysis and discussion of how satellite LAI and snow-depth match observations at flux tower sites is missing.

Line 184-185: Not accounting for the contribution of evaporation to the water vapor flux seems like a limitation of the authors' study.

Line 192-194: Why is this sentence in the section on observed ozone fluxes? I'm not seeing it's relevance.

Line 211-212: So do the authors gap-fill u* at 63 out of 91 sites, or 91 out of 91 sites?

Lines 199-214: Do missing u* measurements correspond to missing energy fluxes? I suspect they might. I also suspect that some of the missing periods may occur during deviations from MOST. Do the authors used gap-filled synthetic fluxes in their analysis, or is the gap-filling just for the dataset just given in the supplemental?

Lines 241-247: This does not make sense to me for the daily estimate. How do the authors pool all the numbers for each hour in a daily estimate when there is only 1-2 numbers for each hour?

Line 247: This sentence is also not clear to me.

Line 251-255: Briefly, will the authors describe the difference between SMA and Sen?

Line 281: What measurements? This transition is a bit abrupt.

Line 289: From equation (3), that the synthetic stomatal ozone flux has little sensitivity to g_ns depends on the relatively low estimate of this value (i.e., stomatal being a large fraction of the total deposition).

Line 302: "stomatal conductance [peaks] when weather conditions favor growth" is quite vague

Lines 303-305: I would say that there is a substantial amount of papers suggesting the exact opposite, and that the references the authors have are quite inappropriate for this statement.

Line 308-310: So the 50% E/ET for Harvard Forest is too low, or are the authors talking about other sites here?

Line 313-316: How do the authors infer these numbers? Are they from Fares et al.? A citation for the seasonality of biogenic emissions is needed. Do we have confidence that this is the seasonal cycle of the BVOCs that matter for ozone fluxes?

Lines 327-328: Is there an "even" missing between "and" and "at"?

Lines 330-343: It is unclear what this section is getting at. The authors find that their synthetic non-stomatal deposition estimate does not match the daily variability in the non-stomatal deposition estimate inferred from observations. Does this necessarily mean that a process is missing, or could it mean that the way the processes are parameterized is wrong? The authors imply the former, but I'm not convinced. I think that the synthetic non-stomatal estimate is not varying in the right way suggests that the synthetic total ozone flux estimate is really limited in its utility.

Lines 354-355: This seems like quite a general statement. I would recommend adding the word "can" in there.

Lines 378-379: How does this "illustrating that ozone . . . far from major industrial emissions" follow from the first part of the sentence? I would only follow this logic if high stomatal conductance is driving high stomatal ozone flux.

Lines 382: Do wetlands have high stomatal conductances inferred from water vapor fluxes because the authors do not account for the evaporation fraction of evapotranspiration?

Line 384-385: That there is the same ranking for the synthetic stomatal flux as the stomatal conductance does not mean that stomata are the main control on ozone deposition, it means they are the main control on stomatal ozone deposition.

Line 389-391: Why? Is this due to stomatal or non-stomatal deposition? If it's due to stomatal deposition, then what does this mean for the ranking of stomatal deposition across land use types?

Line 430: Quantifying differences in spatial variability would be helpful.

Line 464-5: I'm not sure why the second half of this sentence is relevant.

Equations (A6) and (A7) do not follow from Gerosa et al. (2007) equations (5) and (8). I would check them.

---

## Referee Comment (RC2) · Anonymous Referee #2 · 28 May 2018

This manuscript describes the development and application of a method to predict the deposition flux of ozone to vegetation. Specifically, the stomatal deposition is quantified. The method is basd on flux measurements of latent heat (evapotranspiration) and other micrometeorological parameters at over 100 FLUXNET sites in North America and Europe. Measured data are used wherever possible. The analysis is based on the present-day knowledge as available in the literature, and on sound understanding of all processes involved. A rigorous uncertainty analysis is performed. Data of he synthetic flux estimate are compared to measured data of 3 sites. Uncertainties of the synthesic fluxes on the one hand and of the measured fluxes on the other hand were in the same range. Overall, the ozone deposition flux and stomatal flux can be estimated with an error within the factor of 2 or less. Eventually, the potential damage

of the O3 to the vegetation is estimated. It is convincingly shown that the novel dose function is a better proxy than other, widely used proxies such as AOT40 and W126. The manuscript is thoroughly written and very clear. It covers all aspects of ozone deposition to vegetation. It is not only a micrometeorological and physiological analysis, but also reaches out to aspects such as potential damage of vegetation through ozone deposition. It is very interesting to see that the stomatal conductance is a major driver in this regard, likely more than the ozone mixing ratio is. A large portion of the result data set is included in the supplement material. This offers the opportunity to other researchers to make direct use of this data. The only error that this reviewer founds is that in the Appendix, the molar ratio of ozone in units mol mol-1 or ppb is erroneously called concentration. Overall, this is a brilliant manuscript, ready for publication as is (except for the minor error mentioned in the preceding paragraph).

---

## Author Comment (AC1) · 20 Jul 2018

Clifton's comment asks for two things: (1) to clarify our calculation of stomatal and non-stomatal fractions of O3 deposition at Harvard Forest and (2) to provide detailed accounting of the differences in methods and results between this work and that of Clifton et al. (2017). We address both items below and in changes to the manuscript.

While preparing this response, we found an error in the supplemental SynFlux csv file provided with our paper. The files did not include the recalibration of Harvard Forest water vapor flux. No results or figures in the paper were affected, but the gs and other values in that csv file were inconsistent with Figure 5, as Clifton's comment suggested. We have attached corrected csv files to this comment and will ensure that the final pub-

[Figure]

lished paper contains the correct supplemental files. We apologize for this confusion, but very much appreciate that Clifton identified this issue.

In addition, we found that the Harvard Forest data archive (http://harvardforest.fas.harvard.edu:8080/exist/apps/datasets/showData.html?id=hf004), which provides the measured O3 mole fraction and flux measurements used in our work, contained some bad O3 flux values because a data filtering criterion was not applied to the posted data. Filtering out those values caused small changes ($\sim$0.05 cm/s) in the mean deposition velocity of some years. One of us (Munger) is updating the web archive. We will use the updated values in our revisions and in the response below. This change modestly affects the deposition velocity, canopy conductance, and non-stomatal fraction of deposition derived from O3 fluxes at Harvard Forest, so we will update all affected figures.

"Ducker et al. (2018) find that their estimates of stomatal conductance suggest that the nonstomatal fraction of ozone dry deposition ranges from 4 to 32% across years at Harvard Forest and note that this is different from what Clifton et al. (2017) find (20 to 58%) (see Ducker et al. (2018) lines 318-323). Stomatal conductance estimates are critical for inferring the nonstomatal fraction of deposition as the nonstomatal conductance is calculated as a residual of the canopy conductance (inferred from ozone eddy covariance fluxes) and the stomatal conductance. Ducker et al. (2018) attribute the difference between our estimates of the year-to-year range in the nonstomatal fraction of deposition to their re-calibration of the water vapor fluxes used in the stomatal conductance estimate. First, I ask them to clarify whether their range of the nonstomatal fraction is indeed comparable with mine. Second, I use the authors' stomatal conductance estimates (given in their supplementary material) to show below that their estimates of the nonstomatal fraction of deposition are similar to those given by Clifton et al. (2017). I recommend that the authors clarify their discussion of my previous work based on this finding."

Clifton elaborates on these points below, so we respond there.

"First, it is unclear how Ducker et al. (2018) arrive at their reported 4-32% range. When I examine their Figure 5, the figure referenced for these numbers, I can infer the 4-32% range from the error bars, which the caption says represent one standard deviation across daily estimates. If the 4-32% range represents the standard deviation across daily estimates, then these numbers are not directly comparable to mine and the text should be revised accordingly. The 20-58% given in my paper represents the range in the summertime daytime mean nonstomatal conductance across yearly values, not the spread across daily values. If Ducker et al. (2018) actually calculate the mean non-stomatal conductance for each year to obtain their range of 4-32% to compare directly with Clifton et al. (2017), then their approach needs to be more clearly documented in the manuscript."

As suggested, the 4-32% range was not calculated in the same way as Clifton et al. (2017) calculated the interannual variability in non-stomatal deposition, so the two should not be directly compared. We have redone the calculation in the same way as Clifton et al. (2017). This changed our central estimate (from 15% to 9% non-stomatal) and the interannual range. We will replace the text as follows and add supplementary text that provides additional details on the comparison of our results with Clifton et al. (2017). The proposed supplement is provided at the end of our response. The paragraph in Sect. 3.2 (line 318) will become... "A recent analysis of O3 flux measurements at Harvard Forest suggests that non-stomatal deposition averages 40% of daytime O3 deposition during summer months, with a range of 20-60% across years (Clifton et al., 2017). Our analysis of the same site does not support such a large role for non-stomatal deposition at this site in summer. For each year, we calculate summer daytime means of gs and gc by averaging the June-September values, then calculate the non-stomatal fraction of deposition (1- gs/gc). Averaged across years 1993-2000, we find that 8% of daytime O3 deposition is non-stomatal during the summer, with a range of -33% to 34% across years. Negative fractions mean that stomatal conductance is large enough to explain all O3 deposition. A large negative non-stomatal fraction (-33%) occurs in only one year (1996) and no other year is less than -11%, which is within

uncertainty of 0% (2 std.) according to the error propagation. Despite the small or zero non-stomatal fraction found here, our results continue to support the large year-to-year variability of this fraction reported by Clifton et al. (2017). The re-calibrated latent heat flux measurements are the main reason that our results differ from prior work and Supplement S1 provides explanation. At Hyytiälä Forest…[continued as previously]"

We have also revised Figure 5 (see Fig. 1 below) to also show the interannual mean and standard deviation for each month of the year.

"To investigate whether the re-calibration of water vapor fluxes leads to the differences in the fractions of stomatal (or nonstomatal) deposition in Ducker et al. (2018) vs. Clifton et al. (2017), I downloaded the authors' monthly mean stomatal conductance estimates given in their supplementary material. I divided their summertime (June-September) daytime mean stomatal conductance estimates at Harvard by my own estimates of canopy conductance (9am-4pm June-September for each year). My methods for inferring canopy conductance from the ozone eddy covariance flux measurements at Harvard Forest are described in Clifton et al. (2017), and are similar to those of Ducker et al. (2018). Inferring the canopy conductance depends on estimates of the resistances to turbulence and molecular diffusion. These resistances are typically relatively small during the summer daytime compared to the total resistance to deposition calculated from the ozone eddy covariance fluxes, so there should not be substantial differences between our estimates of canopy conductance. Dividing their stomatal conductance estimate by my estimate of canopy conductance suggests that summertime mean stomatal deposition varies from âĹij50 to 100% of the total deposition during 1993- 2000. This corresponds to nonstomatal deposition varying from âĹij0 to 50% of the total ozone dry deposition from year to year (see Figure 1 below). This is wider than the range presented by Ducker et al. (2018) (i.e., 4-32%). Given the uncertainties in the measurements and differences in our approaches especially with respect to inferring stomatal conductance, I think it is fair to say that this range is similar to the range in Clifton et al. (2017) of 20-58%. My analysis here suggests that re-calibrated

water vapor fluxes are not the root cause of the major differences in the ranges given by Ducker et al. (2018) vs. Clifton et al. (2017). Rather, it seems more likely that the differences reflect consideration of the spread in daily variability (Ducker et al., 2018) rather than the year-to-year range (Clifton et al., 2017)."

As we said above, the supplemental files included with our paper did not include the water vapor flux correction at Harvard Forest. As the commenter found, with the uncorrected water vapor fluxes, we estimate roughly the same non-stomatal fraction of O3 deposition as Clifton et al. (2017). However, our best estimate of stomatal conductance at Harvard Forest, using the corrected vapor fluxes, is larger than the numbers used by the commenter, so the stomatal fraction of O3 deposition is larger as well. Again, we apologize for the confusion.

In addition, our monthly and summer averages of gc are generally smaller than those of Clifton et al. (2017). Our hour-by-hour values of gc should be similar, for the reasons given by the commenter, but our monthly averages weight each hourly value by its uncertainty. Large values or outliers of gc tend to have large uncertainty. That is because gc = (vd^-1 - (ra-rb))^-1. When vd^-1 and ra + rb have similar magnitudes, these conditions cause loss of significant figures in subtraction (sometimes called catastrophic cancellation) meaning that gc is very sensitive to measurement errors, which can produce spuriously large gc. Error propagation diagnoses this growth of uncertainty and those values are weighted less in the monthly averages. As a result, our gc values are somewhat smaller (0.4-0.7 cm s-1 for summer averages 1993-2000) and have less interannual variability than Clifton et al. (2017; 0.5-1.2 cm s-1). The lesser variability with error-weighted averages suggests that some apparent interannual variability may be a spurious result of outliers caused by random measurement error. Our weighted gc is similar to the median gc for each month, both of which have similar interannual variability that is less than the unweighted mean gc. Moreover, the unweighted averages are sensitive to how aggressively outliers are discarded, while error-weighted averages are essentially unchanged even if no outliers are excluded.

We will briefly explain the effect of error weighting in Section 2.4 (line 249) by adding, "The error-weighted averages tend to be smaller and less variable than unweighted averages because the error propagation identifies when outliers and large values have greater uncertainty. For example, the monthly values of gc derived from observations at Harvard Forest are $0.57 \pm 0.11$ cm s–1 with error weighting and $0.68 \pm 0.17$ cm s–1 without."

The following paragraph, which will be added to the supplement, addresses the remaining questions about comparing our results with Clifton et al. (2017). – "Our estimate of the non-stomatal fraction of O3 deposition at Harvard Forest (8%, range: -33 to 34%; Sect. 3.2) is smaller than was previously reported at that site (40%, range 20-60%; Clifton et al., 2017). The main reason for the different results is the re-calibration of the water vapor fluxes in this work, which is described in Sect. 2.2. Here, we show how other differences between our analysis and that of Clifton et al. (2017) affect the estimate of non-stomatal fraction of O3 deposition at Harvard Forest. Using our gap-filled data, the annual range of the non-stomatal fraction of O3 deposition at Harvard Forest slightly increases while the mean estimate remains the same (8%, range: -36 to 38%). With uncorrected water vapor fluxes, our estimate would be 51% (range: 32% to 63%). If we also ignore the propagated uncertainty, which varies from hour to hour, and calculate averages with equal weight (i.e. equal uncertainty) for each time interval, as Clifton et al. did, then we would estimate 53% (range: 34% to 66%). If we also use data filtering criteria from Clifton et al. (i.e. remove 3 outliers of vd and gs, but no filtering for precipitation and high relative humidity), then we would estimate 48% (range: 28% to 61%). Finally, if we also restrict our averages to 9am-3pm, as Clifton et al. did, instead of all daylight data, then we would estimate 45% (range: 25% to 60%). This final estimate is very close to the method and value reported by Clifton et al. (2017). The remaining small differences are probably due to Clifton et al. including 1992 in their analysis and differences in the form of the Penman-Monteith and stability functions. Since the re-calibration of water vapor fluxes (Sect. 2.2) is an improvement in this work and the main reason for our results differing from Clifton et al. (2017), our

estimates of small non-stomatal fraction O3 deposition at Harvard Forest appear to be most reliable estimate for this site.

References

Clifton, O. E., Fiore, A. M., Munger, J. W., Malyshev, S., Horowitz, L. W., Shevliakova, E., Paulot, F., Murray, L. T. and Griffin, K. L.: Interannual variability in ozone removal by a temperate deciduous forest, Geophys. Res. Lett., 44(1), 542–552, doi:10.1002/2016GL070923, 2017.

Figure 1. Observed O3 deposition velocity and its in-canopy components at sites with O3 flux measurements. Lines show the multi-year mean and multi-year standard deviation calculated from the monthly averages described in Sect. 2.4. Dashed lines on the stomatal conductance panel show the stomatal fraction of total canopy conductance (gs gcˆ-1) and dashed lines on the non-stomatal conductance panel show the parameterized gns value.

Please also note the supplement to this comment:
https://www.biogeosciences-discuss.net/bg-2018-172/bg-2018-172-AC1-supplement.zip

**O₃ Deposition Velocity (*v_d*)**

**Canopy Conductance (*g_c*)**

**Stomatal Conductance (*g_s*)**

**Non-stomatal Conductance (*g_ns*)**

Blodgett
Hyytiala
Harvard

**Fig. 1.** Revised version of Figure #5

---

## Author Comment (AC2) · 21 Jul 2018

**We appreciate the comments by the reviewer, which have identified areas of the manuscript that can be clarified and strengthened. We have adopted almost all changes suggested by the reviewer, although there are a few points on which we respectfully disagree for reasons that we explain. Our responses are presented in bold.**

**The reviewer suggested some changes in terms and notation for greater clarity. In response, the symbol for synthetic stomatal O3 flux will be $F_{s,O_3}^{\text{syn}}$ and the observation-derived stomatal O3 flux will be $F_{s,O_3}^{\text{obs}}$ We will use these terms in our comments below and throughout the revised manuscript.**

[Figure]

Anonymous Referee 1

Ducker et al. (2018) develop a large dataset of total and stomatal deposition of ozone using micrometeorological observations at flux towers, process models, satellite data, and a gridded product of surface ozone concentrations from air quality networks. They evaluate the simulated deposition against ozone eddy covariance flux observations at three sites with long-term measurements. Then, the authors use their new dataset of simulated stomatal and total deposition to examine the drivers of spatial variability and estimate ozone damage to plants.

The authors harp on the utility of their "synthetic" ozone deposition dataset. Although I see the value in the stomatal deposition estimates, the authors do not really show that their dataset can tell us anything new.

**We respectfully disagree. Sections 3.3 and 3.4 provide several applications of SynFlux: we show and explain the spatial patterns of stomatal O3 uptake on large regional scales; we quantify its comparison to concentration-based metrics on large regional scales; and we assess the range of O3 impacts on vegetation growth. These are new results. We are the first to create a consistent and long-term dataset of stomatal uptake of O3 across a network of sites spanning thousands of kilometers. There is a need for data sets at this scale for ecosystem impact studies and for evaluation of air quality and climate models. The FLUXNET program has shown the value of synthesizing consistent, multi-site datasets of atmosphere-biosphere fluxes and we hope that SynFlux can emulate its success. We are also working on additional applications of SynFlux, which will be published separately because this is already a long paper, and other colleagues have contacted us about using SynFlux in their own work.**

**To better highlight where our new results are found, we will create a new "Section 4. SynFlux applications" and move Sections 3.3 and 3.4 into this section. We will change the name of Section 3 to "SynFlux evaluation".**

Further, I am not convinced that the total ozone deposition estimates are useful, especially when variability in non-stomatal deposition is not simulated accurately. I understand the authors need non-stomatal conductance to estimate the stomatal ozone flux, but is a non-stomatal estimate that gets the variability completely wrong better than a constant? After major revisions I think this manuscript will be suitable for publication.

**And in a related later comment**. . .

I think that the synthetic non-stomatal estimate is not varying in the right way suggests that the synthetic total ozone flux estimate is really limited in its utility.

**We have two responses to these comments. First, the synthetic total O3 flux is not the focus of the paper; the stomatal flux is. Second, and more importantly, we will provide information and revisions that better show the strengths and weaknesses of the non-stomatal parameterization. The situation is somewhat better than was apparent in the manuscript.**

**The synthetic stomatal O3 flux is our marquee product that we highlight in the abstract, conclusions, and throughout the paper, so we think the paper should primarily be judged on its strength. We provide the synthetic total O3 flux as well because it may be useful for *some* purposes, despite the uncertainties that we have documented. As we say in the paper, air quality and climate models often have larger errors (factor of two or greater) in simulated O3 deposition fluxes. Our approach is to provide the critical evaluation so that readers can decide whether the synthetic total O3 flux is useful for their particular applications.**

**We agree that the parameterized non-stomatal conductance has considerable shortcomings, which we quantified and candidly discussed in Sect. 3.2. At the forest sites where we have O3 flux measurements, a constant may be just as good as the parameterization. Nevertheless, SynFlux sites also include crop, shrub, grassland, and wetland sites and the parameterization should perform better than a constant at predicting variations in non-stomatal conductance be-**

tween very different land cover types (Zhang et al., 2002).

We will better discuss the $g_{ns}$ performance at all three sites by replacing the last paragraph of Section 3.2 as follows.

"**The data here in provide an opportunity to evaluate the parameterized non-stomatal conductance (Zhang et al., 2003). The parameterized $g_{ns}$ has similar mean to observation-derived values in summer at Harvard Forest (0.16 vs. 0.12 cm s–1) and Hyytiälä (0.15 vs. 0.25 cm s–1). At Blodgett Forest, the parameterized $g_{ns}$ is about half of observation-derived $g_{ns}$ in summer, but this is not surprising since the parameterization does not account for O3 reactions with biogenic volatile organic compounds (BVOC), which are known to be important at this site (Fares et al., 2010). In winter, however, the parameterized $g_{ns}$ values are similar to observations (0.10 vs. 0.08 cm s–1). The parameterization is therefore able to roughly predict mean non-stomatal conductance in the absence of major BVOC emissions. Nevertheless, the parameterization reproduces almost none of the daily variability of $g_{ns}$ at any site ($R^2$ < 0.1, Fig. R1). This corroborates the recent field assessment that non-stomatal conductance is a weak point of most current dry deposition algorithms (Wu et al., 2018). We attempted, unsuccessfully, to use BVOC emissions from the MEGAN biogenic emission model (Guenther et al., 2012) to improve the $g_{ns}$ parameterization, but the correlations between the daily daytime observation-derived $g_{ns}$ and compounds that react fastest with O3 (monoterpenes and sesquiterpenes) were poor ($R^2 \leq 0.15$). On that basis, $F_{O_3}^{syn}$ may also underestimate total O3 deposition at other sites with high monoterpene and sesquiterpene emissions, such as warm-weather pine forests, but $F_{s,O_3}^{syn}$ should retain its quality everywhere.**"

We will add the following Figure 1 below, which has been referenced in the changes above, to the supplement.

I would like to see a discussion of previous studies that use observed water vapor fluxes

from FLUXNET sites to infer stomatal conductance (e.g., Novick et al., 2016, Lin et al., 2018). I would also like to see a discussion concerning the authors' not accounting for the evaporation contribution to the observed water flux at most sites. There have been several recent papers suggesting partitioning methods or ways of estimating evaporation (e.g., Zhou et al., 2016, Gentine et al, 2016). While for many sites transpiration will dominate the observed water vapor flux during the growing season, evaporation will dominate at other times. I'm not sure why the authors even attempt to estimate deposition during these times.

**We recognize that the quality of stomatal conductance estimates declines at night and outside the growing season, as the evaporative fraction of ET rises. Our intention was to release a continuous SynFlux dataset and allow the user to choose whether or not to use the lower quality data. FLUXNET2015 uses that approach by flagging data with different quality levels. The reviewer suggests a different approach in which only the best data are released. Although we think both approaches have merit, we will adopt the reviewer's suggested approach. We will exclude night and non-growing season data from the paper and supplemental files. The growing season will be defined as days when GPP exceeds 20% of maximum monthly average GPP for that site.**

**This change, and discussion of the suggested papers, will be added to Sect. 2.1 and reflected in changes to the figures and supplementary files. After saying that we calculate stomatal conductance from the inverted Penman-Monteith equation (line 121), we will add, "This method of calculating stomatal conductance has been successfully applied across FLUXNET sites previously (Novick et al., 2016; Knauer et al., 2017; Medlyn et al., 2017; Lin et al., 2018). Those studies and others caution that, since evapotranspiration measurements include evaporation from ground, the stomatal conductance could be overestimated. While there are methods for quantifying the transpiration fraction of evapotranspiration from eddy covariance data (Wang et al., 2014; Zhou et al., 2016; Scott and**

**Biederman, 2017), a more common approach is to restrict analysis to conditions when transpiration dominates. We follow this second approach and use filtering criteria similar to Knauer et al. (2017). We analyze daytime, during the growing season**...."

**Note that we did not discuss the work of Gentine et al. (2016) because it provides a method to estimate total evapotranspiration without flux measurements, but does not address the partitioning issue.**

The difference between the "synthetic" and "observed" stomatal fluxes of ozone needs to be clarified in the manuscript. Do the authors use equation (3) to calculate both? If so, which terms are different between the two calculations? I assume gs is the same between the calculations, so it seems like vd(O3), and gns differ.

**Yes, that is all correct. We provided much of this information on lines 187-192, but we can make that clearer by adding, "Synthetic and observation-derived stomatal O3 fluxes are both calculated with Eq. 3 and use the same observation-derived $g_s$, but use different values of $g_{ns}$, $v_d$, and O3 mole fraction." We will also remind the reader of this distinction in Sect. 3.1.**

The authors should include a discussion of the major drivers of differences between the synthetic and inferred stomatal fluxes in Section 3.1.

**At the end of line 270, we will add, "$F_{s,O_3}^{syn}$ and $F_{s,O_3}^{obs}$ are calculated from the same observation-derived stomatal conductance ($g_s$) and aerodynamic resistances ($r_a$ and $r_b$) but differ in the O3 mole fraction and non-stomatal conductance ($g_{ns}$) that they use (see Sect. 2.1 and 2.2).**

The authors do show the differences in flux-tower and Schnell (O3) in Figure 2, but I think they need to also show and discuss differences in the vd and gns that are from the ozone eddy covariance flux observations vs. estimated with gs and the Zhang model.

**We provided the $g_{ns}$ evaluation in response to an earlier comment and showed**

where this information will be added to Sect. 3.2.

**We will add evaluation of $v_{\mathrm{d}}$ alongside $F_{\mathrm{O}_3}$ in Sect. 3.1 (line 281) and add a figure (Figure 2 below) to the supplement.**

**"The measurements also enable us to evaluate synthetic total deposition, $F_{\mathrm{O}_3}^{\mathrm{syn}}$, and synthetic O3 deposition velocity, $v_{\mathrm{d}}^{\mathrm{syn}}$, although these are less relevant to ecosystem impacts than stomatal uptake, $F_{s,\mathrm{O}_3}^{\mathrm{syn}}$. For daily averages, Figure S5 shows that $F_{\mathrm{O}_3}^{\mathrm{syn}}$ bias (–13 to +65%), slope (0.3-1.4), and $R^2$ (0.05-0.43) are all worse than for $F_{s,\mathrm{O}_3}^{\mathrm{syn}}$. The daily $v_{\mathrm{d}}^{\mathrm{syn}}$ performance is similar (Fig. R2, bias: –26 to +41%, slope: 0.3-1.1, : 0.16-0.37). Monthly averages of $v_{\mathrm{d}}^{\mathrm{syn}}$ and $F_{\mathrm{O}_3}^{\mathrm{syn}}$ both improve the correlation to observations ($R^2$ ˜ 0.12-0.54). The reasons for the better performance of $F_{s,\mathrm{O}_3}^{\mathrm{syn}}$ compared to $F_{\mathrm{O}_3}^{\mathrm{syn}}$ can be derived from Eq. 3, ..."**

**The rest of the paragraph continues as before, until the final sentence, which will also mention $v_{\mathrm{d}}$.**

**"Despite these larger errors, the mean values of $F_{\mathrm{O}_3}^{\mathrm{syn}}$ and $v_{\mathrm{d}}^{\mathrm{syn}}$ are both within 50% of their observed values at two sites and within a factor of 2 at all, which may be useful for some applications, given the paucity of prior $F_{\mathrm{O}_3}$ and $v_{\mathrm{d}}$ measurements."**

The comment from Olivia Clifton needs to be addressed. In general, a clear understanding of how accurate the authors' estimates are on different timescales is critical to understanding the estimates' usefulness. Going back to Clifton et al. 2017, they show strong inter-annual variation in ozone deposition velocity at the Harvard Forest. Do the authors' estimates capture this variation? What about at Blodgett and Hyytiälä?

**Please see our separate response to Clifton. As discussed there, we do observe inter-annual variation, but it is somewhat less than Clifton et al. (2017) reported.**

**We will add at line 328, "If we calculate the summer daytime average of $v_{\mathrm{d}}$ for each year, the interannual range is 0.40-0.68 cm s-1 at Harvard Forest, 0.42-0.65**

**cm s-1 at Blodgett Forest, and 0.43-0.51 cm s-1 at Hyytiälä. This range at Harvard Forest is comparable to other work (0.5-1.2 cm s-1 Clifton et al., 2017), but slightly smaller and less variable because of the error-weighted averages (Sect. 2.4)."**

How much of the variability that is captured with the authors' daily estimates is due to capturing the seasonality of vegetation (i.e., LAI, drought) rather than inter-annual and daily variability?

**$F_{s,O_3}^{\text{syn}}$ has reasonably good skill in predicting daily variability independent of the seasonal cycle. We will add the following information on line 277. "The performance of daily $F_{s,O_3}^{\text{syn}}$ is partially due to resolving the seasonal cycle. If we subtract the mean seasonal cycle from both synthetic and observation-derived daily $F_{s,O_3}$, the residual correlation is $R^2$ = 0.5-0.7 (versus 0.9 with seasonal cycle included). This represents the skill of SynFlux at reproducing within-month and interannual variability."**

Terminology 1) I find the abbreviations used in this study to be very un-intuitive. In the very least I ask that the authors change Fs,O3 to Fsto,O3 so that the "s" can't be confused with "synthetic".

**Since $g_{\text{s}}$ and $g_{\text{ns}}$ are the conventional symbols for stomatal and non-stomatal conductance used in past literature, we will continue to use "s" for stomatal. For "synthetic" variables we will use superscript "syn," as in $F_{s,O_3}^{\text{syn}}$ for synthetic stomatal O3 flux. Observation derived stomatal O3 flux will be represented as $F_{s,O_3}^{\text{syn}}$.**

2) The stomatal conductance, stomatal ozone fluxes, and gns should never be referred to as "observed". I understand that the authors need to distinguish between their synthetic fluxes and the quantities that are inferred from ozone eddy covariance flux observations, but it is misleading to call the latter "observed".

**The reviewer is correct that we used the term "observed" as shorthand to dis-**

tinguish these quantities from "synthetic." We will use the term "observation-derived" instead of "observed" throughout the paper to describe the variables derived from O3 flux measurements.

3) The units for stomatal fluxes are given as nmol m-2 s-1. Is this nmol O3 m-2 s-1?

**Correct. We will use this notation throughout the manuscript.**

Line-by-line comments from anonymous referee 1

Line 67: It's not exactly true that deposition removes 20% of tropospheric ozone. Deposition is 20% of the total loss.

**We will make this change.**

Line 71-73: The authors should elaborate here on the types of ambient reactions that matter for measured ozone fluxes. For example, does reaction of isoprene and ozone in the canopy matter?

**We will add "particularly terpenoid compounds" in this sentence. The paper cited on those lines (Kurpius and Goldstein, 2003) provides further analysis of the in-canopy chemistry, as does our Section 3.2.**

Line 92: I think "minimal additional information from remote sensing and models" is a stretch. The entire synthetic non-stomatal estimate is from a model that relies on remote sensing. If non-stomatal deposition were a minimal fractional of the total, the phrasing would be ok. However, numerous studies have suggested it is not.

**Our meaning was unclear. We meant that we have derived as much information as we can from surface observations and, thus, reduced our dependence on remote sensing and models as much as possible. We will change "minimal" to "some".**

Line 118-119: Again, I take issue with this statement. Yes, the authors' work uses stomatal conductance inferred from observations, which may be better than parameterized stomatal conductance, but the authors are also reliant on modeling and standard meteorology observations for their non-stomatal deposition estimates.

**We meant that we are using *as much information as possible* from observations, but, of course, we are still using some additional assumptions. The reviewer seems to agree that using stomatal conductance inferred from observations is an improvement over past approaches that parameterized stomatal conductance. We will revise this sentence to, "SynFlux aims to constrain O3 deposition and stomatal uptake as much as possible from measured water, heat and momentum fluxes, in contrast to other methods that rely more heavily on atmospheric models or stomatal conductance parameterizations.".**

Line 127-130: I find the authors' argument against using GPP to indicate gs a bit flawed. First, not all GPP-based gs estimates predict gs as a linear function of GPP. Second, the authors are not incorporating nighttime gs into this study, so why does the point about nighttime GPP matter?

**The specific studies that we cited (Lamaud et al., 2009; El-Madany et al., 2017) do assume that $g_s$ is a linear function of GPP, but we see the reviewer's point. We will revise this statement to, "Some studies instead calculate $g_s$ from gross primary productivity (Lamaud et al., 2009; El-Madany et al., 2017), but that method is less widely accepted than the Penman-Monteith approach adopted here."**

Line 147: To my knowledge, the Zhang et al. (2002) parameterization has not been evaluated at sites in North America. Rather Zhang et al. (2002) build their model using ozone fluxes from a couple of sites in the eastern US. This non-stomatal parameterization is rather uncertain (eg., see discussions in Wolfe et al., 2011, Stella et al., 2011, Altimir et al., 2006).

**While the eastern US is, of course, within North America, we see the reviewer's point that the sentence previously implied that the parameterization has been evaluated more extensively than it actually has been. We will revise the sentence,**

so "has been evaluated at sites in North America" becomes "was developed from measurements in the eastern United States."

**We already discuss the accuracy of this parameterization in Sect. 3.2 and we will expand that analysis in response to a later comment. At the end of this paragraph, we will add, "Performance of this non-stomatal parameterization is examined in Sect. 3.2."**

Line 148-150: A brief analysis and discussion of how satellite LAI and snow-depth match observations at flux tower sites is missing.

**We will add here, "Uncertainties in these variables are described in Section 2.4."**

**In Section 2.4 (line 233), we will expand our description of LAI and snow-depth uncertainties. "For remotely sensed leaf area index, the uncertainty is 1.1 m2 m-2 for all vegetation types (Claverie et al., 2013; 2016). Snow depth uncertainty in MERRA2 is 0.08 m (Reichle et al., 2017)."**

Line 184-185: Not accounting for the contribution of evaporation to the water vapor flux seems like a limitation of the authors' study.

**See our response to the earlier comment about partitioning of evapotranspiration.**

Line 192-194: Why is this sentence in the section on observed ozone fluxes? I'm not seeing its relevance.

**The outliers in the synthetic and inferred stomatal ozone fluxes are due to high uncertainties within the heat and water vapor fluxes that were reported in the FLUXNET2015 dataset. However, we agree that this information is probably more useful within section 3.1, so we will move the sentence into that section (line 277).**

Line 211-212: So do the authors gap-fill u* at 63 out of 91 sites, or 91 out of 91 sites?

**We previously used u\* gap-filling at all sites, however, as the reviewer suggests, this is probably unhelpful at sites with low $R^2$. In the revised paper and dataset, we will only use u\* gap filling at sites with $R^2$ > 0.5, which will be stated within lines 211-212. As a result, the gap-filled SynFlux values will slightly change at some sites. The revised sentences will say: "The predicted friction velocities from the regression model are correlated with available observations ($R^2$ > 0.5) and have minimal mean bias ($\pm 10\%$) at 85 out of 91 eligible sites (Fig. S3), with most sites (63 out of 91) showing strong correlations ($R^2$ > 0.7). At the remaining 6 sites with lower regression model performance ($R^2$. < 0.5) we do not use u\* gap-filling."**

Lines 199-214: Do missing u\* measurements correspond to missing energy fluxes? I suspect they might. I also suspect that some of the missing periods may occur during deviations from MOST. Do the authors used gap-filled synthetic fluxes in their analysis, or is the gap-filling just for the dataset just given in the supplemental?

**Yes, missing u\* measurements occur at the same time that energy fluxes are missing or already gap-filled by the FLUXNET team. These may be missing due to unsuitable atmospheric conditions, including fog and rain, or equipment problems and maintenance, so missing u\* doesn't necessarily imply deviations from MOST. To clarify this issue, we will add in the manuscript (line 212) the following: "Time periods with u\* gaps have no significant bias in meteorological conditions (e.g. mean wind speed, radiation, energy fluxes) compared to periods with u\* measurements. As a result, the differences in monthly mean $F_{s,O_3}^{\text{syn}}$ with and without gap filling are 10% (rms). So, although the u\* gap filling is a potential source of uncertainty, the $F_{s,O_3}^{\text{syn}}$ estimates are robust. The following analysis will use the gap-filled data, but our results do not change in any meaningful way if we use the unfilled data."**

Lines 241-247: This does not make sense to me for the daily estimate. How do the authors pool all the numbers for each hour in a daily estimate when there is only 1-2

numbers for each hour?

**Correct. There are two observations per hour at almost all sites (one per hour at a few sites). We pool these with a maximum likelihood estimate, whose formula is provided in Appendix B, which is cited in this paragraph. The method is well-defined for any number of values, including 1-2. In cases with only 1 number, the mean and uncertainty for that hour is simply the value and uncertainty of the one available value. To make this more clear, we will add sentences (line 246), "For the daily averages, there are 1-2 observations within each hour. For the monthly averages, there are typically 30-60 in each hour of the day."**

Line 247: This sentence is also not clear to me.

**We will revise the sentence, "We calculate seasonal averages with an un-weighted mean of monthly values."**

Line 251-255: Briefly, will the authors describe the difference between SMA and Sen?

**SMA is a parametric slope estimator while Thiel-Sen is a non-parametric slope estimator. SMA is therefore relatively more efficient while Thiel-Sen is more robust against outliers. We will change this sentence to say, "We quantify linear relationships between variables using a parametric method (standard major axis or SMA, Warton et al. 2006) and a non-parametric method (Thiel-Sen slope, Sen, 1968), which is more robust against outliers."**

Line 281: What measurements? This transition is a bit abrupt.

**We are referring to the O3 flux measurements. We will revise this sentence to, "Measurements of total O3 flux enable us to evaluate synthetic total deposition, $F_{O_3}^{syn}$, as well, although this is less relevant to ecosystem impacts than stomatal uptake, $F_{s,O_3}^{syn}$."**

Line 289: From equation (3), that the synthetic stomatal ozone flux has little sensitivity to gns depends on the relatively low estimate of this value (i.e., stomatal being a large

fraction of the total deposition).

**Actually, no. The derivation in this paragraph assumes nothing about the relative magnitudes of $g_s$ and $g_{ns}$. The only assumption is that $v_d \approx g_c$, which is accurate most of the time (except under very stable atmospheric conditions). As a result, $F_{s,O_3}^{syn}$ has little sensitivity to $g_{ns}$ regardless of whether stomatal or non-stomatal conductance is larger. The stomatal O3 flux becomes sensitive to $g_{ns}$ only when aerodynamic or quasi-laminar resistance rivals canopy conductance. We will add sentences noting that, "$F_{s,O_3}^{syn}$ has little sensitivity to $g_{ns}$ regardless of whether stomatal or non-stomatal conductance is larger. We confirm this insensitivity in tests where the parameterized $g_{ns}$ value is doubled at ten sites. The hourly $F_{s,O_3}^{syn}$ values change only 3-8%."**

Line 302: "stomatal conductance (peaks) when weather conditions favor growth" is quite vague

**We will change this to "stomatal conductance [peaks] during warm and wet months".**

Lines 303-305: I would say that there is a substantial amount of papers suggesting the exact opposite, and that the references the authors have are quite inappropriate for this statement.

**We meant this more as a statement of "conventional wisdom." For decades, most atmospheric scientists have generally thought that stomatal conductance is generally larger than non-stomatal conductance for O3, based on the influential parameterizations by Wesely (1989) and Zhang et al. (2003). However, the reviewer is correct that recent papers have challenged this conventional wisdom. We will restate our sentence in a historical context: "Traditionally, stomatal conductance was thought to exceed non-stomatal conductance during the growing season at most vegetated sites (Wesely, 1989; Zhang et al., 2003), although this has been challenged more recently (Altimir et al., 2006; Stella et al., 2011; Wolfe**

**et al., 2011; Plake et al., 2015).''**

Line 308-310: So the 50% E/ET for Harvard Forest is too low, or are the authors talking about other sites here?

**Yes, the 50% E/ET was probably too low. However, the point is moot because, as explained above, we now exclude months when the biosphere is mostly dormant. We will remove discussion of the winter seasons at Harvard and Hyytiälä Forests.**

Line 313-316: How do the authors infer these numbers? Are they from Fares et al.? A citation for the seasonality of biogenic emissions is needed. Do we have confidence that this is the seasonal cycle of the BVOCs that matter for ozone fluxes?

**Yes, this information is from literature. We will add ''as documented in past work'' and also cite the work of Wolfe et al. (2011). The prior studies that we have cited already in this sentence addressed the reviewer's questions about BVOC seasonality and O3 reactivity at Blodgett Forest (Kurpuis and Goldstein, 2003; Fares et al., 2010; Wolfe et al. 2011), although there are certainly unresolved details about BVOC emissions and O3-BVOC chemistry.**

Lines 327-328: Is there an ''even'' missing between ''and'' and ''at''?

**Yes. We will fix this.**

Lines 330-343: It is unclear what this section is getting at.

**Our goal in this paragraph is to evaluate the non-stomatal parameterization because, as the reviewer noted in an earlier comment, the community needs more field evaluation of these parameterizations. We will add an introductory statement (line 330), ''The data here provide an opportunity to evaluate the parameterized non-stomatal conductance predicted by the parameterization.'' Additional changes in this paragraph, described above, should also clarify this paragraph.**

The authors find that their synthetic non-stomatal deposition estimate does not match

the daily variability in the non-stomatal deposition estimate inferred from observations. Does this necessarily mean that a process is missing, or could it mean that the way the processes are parameterized is wrong? The authors imply the former, but I'm not convinced.

**This is a helpful clarification. We will add that a process is "misrepresented or missing".**

Lines 354-355: This seems like quite a general statement. I would recommend adding the word "can" in there.

**We will do this.**

Lines 378-379: How does this "illustrating that ozone . . . far from major industrial emissions" follow from the first part of the sentence? I would only follow this logic if high stomatal conductance is driving high stomatal ozone flux.

**Correct. That is what we meant. To clarify, we will rewrite, "illustrating that O3 can impact remote ecosystems with high stomatal conductance, even where O3 concentrations are low."**

Lines 382: Do wetlands have high stomatal conductances inferred from water vapor fluxes because the authors do not account for the evaporation fraction of evapotranspiration?

**Yes, evaporation could be a confounding factor, although several of these wetland sites have dense vegetation covering the water, which reduces the evaporative fraction of evapotranspiration. The inferred stomatal conductance at wetland sites ($0.48 \pm 0.16$ cm/s) is also within a reasonable range for wetland vegetation (e.g. up to 1 cm/s in Drake et al., 2013). We will add a caveat here and in a footnote to Table 2 noting that evaporation at wetland sites could result in overestimating stomatal conductance.**

Line 384-385: That there is the same ranking for the synthetic stomatal flux as the

stomatal conductance does not mean that stomata are the main control on ozone deposition, it means they are the main control on stomatal ozone deposition.

**Thank you for this correction. The sentence will be rewritten as: "The vegetation types rank in the same order for stomatal conductance, again showing stomata as the main control on O3 uptake into vegetation.".**

Line 389-391: Why? Is this due to stomatal or non-stomatal deposition? If it's due to stomatal deposition, then what does this mean for the ranking of stomatal deposition across land use types?

**This is a good question. However, Silva and Heald (2017) did not provide information on stomatal and non-stomatal pathways necessary to provide an answer.**

Line 430: Quantifying differences in spatial variability would be helpful.

**We will provide the spatial correlation coefficients of the various concentration-based metrics. The statement will be revised, "AOT40 and W126 are well correlated with each other across sites ($R^2$ = 0.87) and with mean O3 mole fraction ($R^2$ = 0.76 and $R^2$ = 0.52 for mean O3 vs. AOT40 and W126, respectively) despite their different weighting functions. As a result, all of these concentrations-based metrics have similar spatial patterns in the US and Europe."**

Line 464-5: I'm not sure why the second half of this sentence is relevant.

**This sentence was somewhat redundant with an earlier sentence in the same paragraph. We will move the citation earlier in the paragraph ("Although species vary in their sensitivity to O3 (e.g. Lombardozzi et al., 2013)...") and delete the sentence.**

Equations (A6) and (A7) do not follow from Gerosa et al. (2007) equations (5) and (8). I would check them.

**We have checked that the equations are correct as written and derived from**

Gerosa et al. (2007), with two minor differences. We express the water vapor flux in terms of vapor pressure, *e*, instead of specific humidity, $q = \epsilon e / p$, but this is a basic meteorological identity. There is also a sign change because we define heat flow from the surface to the atmosphere as positive flux, which we say in the Appendix, while Gerosa et al. define it as negative flux. The equations are otherwise equivalent, so we don't think any changes are needed.

[revised manuscript text omitted]

---

## Author Comment (AC3) · 21 Jul 2018

This manuscript describes the development and application of a method to predict the deposition flux of ozone to vegetation. Specifically, the stomatal deposition is quantified. The method is based on flux measurements of latent heat (evapotranspiration) and other micrometeorological parameters at over 100 FLUXNET sites in North America and Europe. Measured data are used wherever possible. The analysis is based on the present-day knowledge as available in the literature, and on sound understanding of all processes involved. A rigorous uncertainty analysis is performed. Data of the synthetic flux estimate are compared to measured data of 3 sites. Uncertainties of the synthetic fluxes on the one hand and of the measured fluxes on the other hand were in the same range. Overall, the ozone deposition flux and stomatal flux can be

estimated with an error within the factor of 2 or less. Eventually, the potential damage of the O3 to the vegetation is estimated. It is convincingly shown that the novel dose function is a better proxy than other, widely used proxies such as AOT40 and W126. The manuscript is thoroughly written and very clear. It covers all aspects of ozone deposition to vegetation. It is not only a micrometeorological and physiological analysis, but also reaches out to aspects such as potential damage of vegetation through ozone deposition. It is very interesting to see that the stomatal conductance is a major driver in this regard, likely more than the ozone mixing ratio is. A large portion of the result data set is included in the supplement material. This offers the opportunity to other researchers to make direct use of this data. The only error that this reviewer founds is that in the Appendix, the molar ratio of ozone in units mol mol-1 or ppb is erroneously called concentration. Overall, this is a brilliant manuscript, ready for publication as is (except for the minor error mentioned in the preceding paragraph).

**Thank you for pointing this issue out within Appendix A and your comments about our manuscript. We will fix the description of O3 mole fraction within the Appendix.**